REGISTERED REPORT PROTOCOL

# Age-preserved semantic memory and the CRUNCH effect manifested as differential semantic control networks: An fMRI study

Niobe Haitas [1,2]*, Mahnoush Amiri[1], Maximiliano Wilson[3], Yves Joanette[1,2], Jason Steffener[4]

**1** Laboratory of Communication and Aging, Institut Universitaire de Gériatrie de Montréal, Montreal, Quebec, Canada, **2** Faculty of Medicine, University of Montreal, Montreal, Quebec, Canada, **3** Centre de Recherche CERVO – CIUSSS de la Capitale-Nationale et Département de Réadaptation, Université Laval, Quebec City, Quebec, Canada, **4** Interdisciplinary School of Health Sciences, University of Ottawa, Ottawa, Ontario, Canada

* niobe.haitas@gmail.com

This is a Registered Report and may have an associated publication; please check the article page on the journal site for any related articles.

## Abstract

Semantic memory representations are overall well-maintained in aging whereas semantic control is thought to be more affected. To explain this phenomenon, this study aims to test the predictions of the Compensation Related Utilization of Neural Circuits Hypothesis (CRUNCH) focusing on task demands in aging as a possible framework. The CRUNCH effect would manifest itself in semantic tasks through a compensatory increase in neural activation in semantic control network regions but only up to a certain threshold of task demands. This study will compare 40 young (20–35 years old) with 40 older participants (60–75 years old) in a triad-based semantic judgment task performed in an fMRI scanner while manipulating levels of task demands (low vs. high) through semantic distance. In line with the CRUNCH predictions, differences in neurofunctional activation and behavioral performance (accuracy and response times) are expected in young vs. old participants in the low- vs. high-demand conditions manifested in semantic control Regions of Interest.

## Introduction

Language overall is well preserved in aging [1] and semantic memory may even improve across the lifespan [2–6], despite numerous neurophysiological declines in other cognitive domains that occur in the aging brain [6–8]. When compared with attention or memory, the relative preservation of language throughout lifetime [9] could be justified by the necessity to maintain successful communication, resulting in compensatory, flexible or atypical recruitment of neural resources [6]. Performance in terms of accuracy in semantic tasks is generally well maintained in older adults considering their more extensive experience with word use and a larger vocabulary than younger adults [2, 5, 6, 10–13]. Response times (RTs) however are often longer compared to younger adults [10], possibly because older adults are slower in accessing and retrieving conceptual representations from their semantic store [14–16], engaging the required

**Data Availability Statement:** All related data (stimuli, instructions, ethics' approval) are in the osf.io platform (doi: 10.17605/OSF.IO/F2XW9) https://osf.io/f2xw9/?view_only= c36d4ac68e6d422ba0208ff2eda617bc. In addition, once they become available, we will upload our unthresholded statistical maps to neurovault (https://neurovault.org/), an online platform sharing activation data. Permanent links to the unthresholded statistical maps to be uploaded at Neurovault will be provided as part of the dataset deposited on the OSF, under the same DOI (DOI: 10.17605/OSF.IO/F2XW9). Though the authors intend to make their raw data publicly available, ethical regulations at our institute do not allow for sharing of raw data at the moment, due to privacy risks for the human subjects and risk of re-identification (data contain potentially identifying information). These will remain stored in a private server, accessible on demand and following ethics committee approval. Data access requests can be made at: Unité de Neuroimagerie Fonctionnelle (UNF) https://unf-montreal.ca/contact/ Centre de Recherche de l'institut Universitaire de Gériatrie de Montréal 4565 Queen-Mary Road.

**Funding:** The study was funded by the Canadian Institutes of Health Research (CIHR).

**Competing interests:** The authors have declared that no competing interests exist.

executive function resources [17], and necessary motor responses [18]. Aside from behavioral performance, findings reported in the literature about the neural correlates sustaining semantic memory of older adults when compared with younger ones, are often conflicting, depending on the task utilized, inter-individual variability and the specific age group. Though several age-focused neurofunctional reorganization phenomena (e.g. Hemispheric Asymmetry Reduction in Older Adults-HAROLD [19] and Posterior-Anterior Shift in Aging-PASA [20] aim to explain how aging affects cognitive skills in general, it is still not clear how aging impacts the underlying pattern of activation sustaining semantic memory, given its relative life-course preservation. The relative preservation of semantic memory performance in older adults when compared with other cognitive fields [9, 21, 22] could be partly justified by the proposed dual nature of the semantic memory system, as expressed within the controlled semantic cognition framework [23–26]. The present study focuses on the question of preservation of semantic memory in aging, defined as the 'cognitive act of accessing stored knowledge about the world' [27] using a semantic judgment task manipulating semantic control with two demand levels (low and high).

To account for conflicting findings in terms of brain activation during semantic memory tasks and the relative preservation of semantic memory in normal aging, a possible explanation is to consider it the result of adaptive mechanisms captured within the CRUNCH model (Compensation Related Utilization of Neural Circuits Hypothesis) [28]. This theory states that it is the level of task difficulty that impacts performance and neurofunctional activation in both younger and older individuals, whereas aging could be thought of as the expression of increasing task demands earlier than in younger adults. Accordingly, additional neural resources are recruited to attempt compensation when faced with elevated task requirements, echoing an aspect of the aging process manifestation [29, 30]. Compensation is thus defined as 'the cognition-enhancing recruitment of neural resources in response to relatively high cognitive demand' [29]. Alternatively, age-related reorganization phenomena refer to reduced neural efficiency, also known as dedifferentiation, resulting in reduced performance in the old [31–34].

At the same time and as part of the age-related neurofunctional reorganization, neural resources may migrate from the default mode network (DMN) towards more urgent task requirements, which can be expressed as underactivation in such areas subserving 'redundant' tasks [28]. Indeed, the more task demands increase, the more DMN activation is expected to decrease, however this ability to 'silence' the DMN reduces in older adults [35]. Both over- and underactivation are relevant terms referring to comparisons with optimal patterns of activation as seen in younger adults [28]. Although the CRUNCH model describes compensatory neural mechanisms, it is not without its limits. For older adults, the overactivation benefit is thought to reach a threshold beyond which additional neural resources do not suffice, after which activation declines and performance deteriorates [28]. The relationship between task demands and fMRI activation has been described as an inverted U-shaped one, with the curve of older adults being to the left of the curve of younger ones. In other words, older adults would recruit additional neural resources at lower levels of task demands, reach a maximum and decrease in activity as task demands continue to increase earlier than younger ones (see Fig 3a in [29, 30]).

The CRUNCH hypothesis was conceived on evidence from a working memory study. Activation increased in the dorsolateral prefrontal cortex when accuracy was maintained and decreased when accuracy was compromised, depending on task load, or else, the number of items successfully retained [36, 37]. Congruent results were found in another working memory study, claiming that older adults may achieve the same outcomes using different neural circuits or strategies to achieve age-matched performance [38]. However, the CRUNCH predictions

were not confirmed in recent working memory studies. In a working memory study with 3 load conditions using functional near-infrared spectroscopy (fNIRS), activation in the young progressively increased in the PFC as difficulty increased and performance was maintained [39]. However in the older adults, when performance was compromised during the most difficult condition, activation in the PFC bilaterally remained high. Similarly, in a visuospatial working memory task with 4 levels of task demands, the CRUNCH predictions were not found [40]. Instead, an increase in activation was found in a large network (premotor, prefrontal, subcortical and visual regions) however, no 'crunch' point after which activation decreases was found for the older group. Though older adults showed increased activation across regions at the higher task loads when compared with the younger ones, at the group level this difference was not significant, thus challenging the CRUNCH prediction of interaction between difficulty and fMRI activation.

Compatible with the CRUNCH expectations, increased activations with relatively maintained performance have been reported in frontoparietal regions in several language studies, however the results are not always consistent. More precisely, in a discourse comprehension study using fNIRS, increased activation was found in the left dorsolateral prefrontal cortex in older adults while performance was mostly equal to their younger counterparts [41]. In a sentence comprehension study, increased activation was observed in both younger and older adults during the more complex sentences in regions such as the bilateral ventral inferior frontal gyrus (IFG)/anterior insula, bilateral middle frontal gyrus (MFG), bilateral middle temporal gyrus (MTG), and left inferior parietal lobe [42]. Older adults showed increased activity compared with the young in the IFG bilaterally and the anterior insula in the difficult condition, however their performance in terms of accuracy was not maintained. Partially compatible with CRUNCH, overactivations with maintained performance have also been observed in a picture naming study manipulating for task demands/inhibition [43]. When naming difficulty increased, both younger and older adults showed increased activation in bilateral regions such as the IFG, the anterior cingulate gyri, the pre-, post-central, supramarginal and angular gyri, together with maintained performance while response times (RTs) of older adults did not significantly increase [43]. Fewer studies exist on semantic memory in light of increasing task demands, which is the focus of the current study.

Given the large volume of concepts and processes involved, semantic memory relies on a widely distributed and interconnected mainly left-lateralized core semantic network [17, 27, 44–46] and bilaterally the anterior temporal lobes (ATL) proposed to act as semantic hubs [47, 48]. Semantic memory is suggested to be organized as a dual system composed of two distinctive but interacting systems, one specific to representations and one specific to cognitive-semantic control [25, 46, 49–53]. In other words, it is thought to include processes related to stored concept representations with their modality-specific features which would interact with control processes in charge of selecting, retrieving, manipulating and monitoring them for relevance and the specific context, while at the same time suppressing irrelevant information [24–26, 54–57]. Within the controlled semantic cognition framework [26], the semantic control network would be significantly recruited during more complex tasks underpinned by left-hemisphere regions such as the prefrontal cortex (PFC), inferior frontal gyrus (IFG), posterior middle temporal gyrus (pMTG), dorsal angular gyrus (dAG), dorsal anterior cingulate (dACC), and dorsal inferior parietal cortex (dIPC) [25, 26, 45, 46, 51, 53, 58, 59], potentially extending towards the right IFG and PFC when demands further intensify [46]. One of the most up-to-date and extensive meta-analysis of 925 peaks over 126 contrasts from 87 studies on semantic control and 257 on semantic memory, found further evidence for the regions involved in semantic control, concluding them to the left-lateralized IFG, pMTG, pITG (posterior inferior temporal gyrus), and dmPFC (dorsomedial prefrontal cortex) regions [24].

Regions related to semantic control are thought to be largely overlapped by the neural correlates of the semantic network [24] but also thought to largely overlap with regions related to the 'multiple-demand' frontoparietal cognitive control network involved in planning and regulating cognitive processes [26, 60].

Differential recruitment has been found for easy and harder semantic tasks in young adults including recruitment of semantic control regions for the latter. In a study using transcranial magnetic stimulation (TMS) on the roles of the angular gyrus (AG) and the pMTG, participants were required to perform identity or thematic matchings that were either strongly or weakly associated, based on ratings previously collected and where RTs were used as a function of association strength. Stimulation to the AG and the pMTG confirmed their roles in more automatic and more controlled retrieval respectively [58]. An fMRI study used a triad-based semantic similarity judgment task to compare between concrete and abstract nouns (imageability) while manipulating additionally for difficulty. Difficulty was based on semantic similarity scores based on ratings of words, and for every triad, a semantic similarity score was computed to classify them as easy or hard. Increased activations were found during the hard triads and regardless of word imageability, in regions modulating attention and response monitoring such as bilaterally in the cingulate sulcus, the medial superior frontal gyrus and left dorsal inferior frontal gyrus [61]. In a triad-based synonym judgment task comparing concrete vs. abstract words, where triads were categorized as easy or difficult based on the respective response time in relation to the group mean, a main effect of difficulty was confirmed, with increased activations reported in the left temporal pole, left IFG and left MTG [62]. In a triad-based task where participants were requested to match words for colour and semantic relation to probe more automatic or controlled semantic processing respectively, greater activation was found in the IFG and IPS during the more difficult triads that were based on colour-matching. Accuracy was overall maintained equally across conditions but there were more errors and longer RTs in the 'difficult' colour condition, lending support to the controlled semantic cognition idea [50]. There is therefore evidence to support an increase in activation in semantic control regions when semantic processing demands increase, which could be attributed to 'matching' task requirements with available neural resources, in line with CRUNCH predictions. When it comes to aging, though the system related to representations is thought to be well-maintained, the system related to cognitive-semantic control is thought to be more affected [23]. This study focuses on how the relation between semantic control network activation and increasing task demands is affected by aging.

The neural correlates sustaining semantic memory are thought to be largely age-invariant, with only small differences existing in neural recruitment as a function of age [16, 22, 63–66]. In a recently conducted meta-analysis of 47 neuroimaging studies comparing younger and older people, increases in activation in semantic control regions in older adults were reported when compared with younger ones, while accuracy was found to be equal between the two groups [22]. Though this increase in activation could be attributed to compensatory accounts, it could also reflect age-related loss of neuronal specificity or efficiency [22]. Several studies report activation and performance results in line with the compensatory overactivation account. In a semantic judgment task, participants had to decide whether two words share a common feature (shape or color) with their performance being categorized as better or worse based on a split from behavioral data [56]. In better performing older adults, activation was increased relative to younger adults in control regions such as the inferior parietal and bilateral premotor cortex, regions important for executive functions and object visual processing as well as relative to poorer performing older adults, in the premotor, inferior parietal and lateral occipital cortex. A further analysis for gray matter found that increased gray matter in the right precentral gyrus was associated with maintained performance [56]. In a semantic

categorization study, older participants performed as accurately as the younger ones but had slower RTs. Their maintained performance was correlated with activation in a larger network than the one of younger ones, including parts of the semantic control network (such as left frontal and superior parietal cortex, left anterior cingulate, right angular gyrus and right superior temporal cortex), which was reportedly atypical and excluded the PFC [44].

Specifically to left IFG recruitment, believed to be in charge of top-down semantic control [45, 49, 51, 67], its association with the 'difficult' condition has been reported in several studies. In a triad-based semantic judgment task evaluating for rhyme, semantic and perceptual similarity, interaction and conjunction analyses revealed a significant interaction between age and the high-load semantic condition. Older adults overrecruited the control-related regions of the left IFG, left fusiform gyrus and posterior cingulate bilaterally, when competition demands increased while their accuracy was even better than their younger counterparts [66]. In a picture-naming task, older adults recruited overall larger frontal areas than younger ones in both hemispheres. Though the bilateral -and not the solely-left- recruitment of the IFG was beneficial to performance of older participants, the recruitment of other right-hemisphere regions was negatively correlated with accuracy [16]. The authors provided support to the finding that the neural substrates for semantic memory representations are intact in older adults whereas it is the executive aspect of language functions, including accessing and manipulating verbal information, that are most affected by aging [16]. In another study with young adults only, aiming to dissociate the role of the IFG in phonologically vs. semantically cued word retrieval, the recruitment of anterior-dorsal parts of the LIFG was associated with the high task demands condition in the semantic fluency condition, while performance was maintained [68].

Evidence therefore exists for a correlation between an increase in activation of semantic control regions when faced with increased task demands, which could be indicative of the compensation account to favor semantic memory performance in both young and older adults, and potentially reflecting the ascending part of the U-shaped relation between fMRI activation and task demands. Attributing however a causal relation between increased activation in the semantic control network and compensation is not straightforward. Distinguishing between the compensation and de-differentiation accounts can be challenging, as merely correlating brain activation with behavioral outcomes to claim compensation is methodologically incomplete [69, 70]. Many studies do not manipulate or cannot be compared for task demands and thus interpreting results that correlate neural activation with behavior can be confusing [53]. For example, in a study where task demands are lower, reorganization may be interpreted as compensatory when performance is maintained whereas when performance is more affected, it can be attributed to dedifferentiation. Numerous methodological caveats exist when attempting to allocate meaning a posteriori to age-related reorganization, given the observational nature of neuroscience, but also the need for more robust methodological designs, including longitudinal studies that measure in-person changes, between regions comparison and better analytic approaches (for a review see [70]). Correlating increased activation with improved performance at a single point in time and attributing it to compensation would require additional measures, also given that compensation may be attempted or only partly successful [30, 71].

According to the CRUNCH theory, the compensatory increase in activation of semantic control regions is thought to reach a plateau beyond which additional resources no longer benefit performance [28]. As such, reduced activation in cognitive control regions when semantic processing demands increase has also been reported. According to CRUNCH, this reduced activation could be interpreted as neural resources having already reached their maximum capacity and no longer being sufficient to successfully sustain compensation for the task [28]. Indeed, the meta-analysis of 47 neuroimaging studies comparing activation in young and older adults (mean age of young participants: 26 years (SD = 4.1) and mean age of older participants:

69.1 (SD = 4.7) during semantic processing tasks, also reported decreased activation in the older adults in typical semantic control regions in the left hemisphere (IFG, pMTG, ventral occipitotemporal regions and dIPC) together with increased activation in 'multiple-demand network' regions in the right hemisphere (IFG, right superior frontal and parietal cortex including the middle frontal gyrus, dIPC and dACC) especially when performance was sub-optimal [22]. In a semantic judgment task (living vs. non-living judgement of words) study with two levels of difficulty and four across-the-lifespan age groups, activation outside the core semantic network increased with age linearly and contralaterally towards the right hemisphere (right parietal cortex and middle frontal gyrus) in the easy condition, while accuracy was maintained [64]. In the difficult condition however, RTs were slower and reduced activation was observed in older participants in semantic control regions, namely the frontal, parietal and cingulate cortex regions, suggesting a declining ability of brain to respond to increasing task demands by mobilizing semantic control network resources as age increases [64].

Similarly, increased activation in right-lateralized semantic control regions was detrimental to performance in both young and old participants in a word generation study manipulating for task difficulty [72]. Indeed, activation in the ventral IFG bilaterally was correlated with difficult items as opposed to easier ones and reduced performance irrespective of age. In a verbal fluency study by the same group using correlation analysis, a strong negative correlation was found between performance and activation in the right inferior and middle frontal gyrus ROIs [73]. Older adults demonstrated a more bilateral activation than younger ones especially in the right inferior and middle frontal regions whereas their performance during the semantic task was negatively impacted. However, this right-lateralized semantic control network increase in activation together with a drop in performance has not been consistently documented. For example, in a semantic judgment task on word concreteness using magnetic encephalography (MEG), older participants overactivated the right posterior middle temporal gyrus, inferior parietal lobule, angular gyrus and the left ATL and underactivated the control-related left IPC as a result of increased task demands while their performance was equivalent to the young, thus lending support to compensatory accounts [65]. According to CRUNCH, the above findings could be interpreted within the descending part of the inverted U-shaped relation between semantic processing demands and fMRI activation [29], whereby after a certain difficulty threshold, available neural resources from the semantic or multiple-demand control network have reached their maximum capacity and further lead to reduced activations and a decline in performance [30].

In summary, it seems that depending on the semantic task used and its perceived or actual difficulty, both increased and decreased activations have been reported in the semantic control network along with variations in consequent performances. The relationship between neural activation, task difficulty and behavioral performance is not straightforward. It is possible that the neural correlates of semantic memory remain relatively invariant throughout aging when the task is perceived as easy. On the other hand, when task difficulty or the perception of it increases, activation and behavioral performance may increase or reduce depending on the nature of the task and its level of perceived or actual difficulty, in line with CRUNCH. Accordingly, maintained performance could depend on the additional recruitment of semantic control network resources but only between certain thresholds of difficulty, before which increasing activation would be unnecessary or beneficial and after which performance would decline.

### Age-related reorganization phenomena alternative to CRUNCH

A number of alternative neurofunctional reorganization phenomena have been reported to account for the evolution of general cognitive skills in aging (for reviews, see ([30, 74, 75]).

Such phenomena often refer to the engagement of compensatory mechanisms and redistribution of resources through overactivation or deactivation often including in the PFC [28, 30]. For example, the HAROLD neurofunctional reorganization phenomenon refers to a hemispheric asymmetry reduction in older adults with the objective of maintaining high performance [19]. To reduce the asymmetry, brain activation can increase and/or decrease in certain brain areas by recruiting additional and alternative neuronal circuits from the contralateral hemisphere. The resulting asymmetry reduction optimizes performance, whereas elderly adults who maintain a unilateral or asymmetrical activation pattern similar to the young, do not perform as well [19]. Several studies have recently challenged the accuracy of the HAROLD model [76, 77]. An alternative pattern of neurofunctional reorganization has been reported to occur intrahemispherically. The PASA (Posterior Anterior Shift in Aging) phenomenon provides a picture of such type of reorganization [78], describing an age-related shifting of activation from the occipitotemporal to the frontal cortex [20, 79]. PASA is considered to reflect a general age-related compensation phenomenon for processing sensory deficits by decreasing activation in occipitotemporal regions and increasing activation in frontal regions rather than reflect task difficulty [20]. A recent metaanalysis [80] on healthy aging provided support for the findings of the PASA phenomenon, however, others have challenged its compensatory claim [81]. Additionnally to the above intra- and inter-hemispheric reorganization phenomena is the 'cognitive reserve' hypothesis, which attributes successful cognitive processing in aging to complex interactions between genetic and environmental factors that influence brain reserve and the brain's ability to compensate for age-related pathologies [82]. Cognitive reserve is proposed to depend on both neural reserve and neural compensation, a distinction reflecting inter-individual variability to use resources efficiently, flexibly or differently while performing cognitive tasks but also using alternative strategies in pathological situations. Accordingly, older adults can adapt to aging and cope with increased task demands in a flexible manner by activating regions similarly to the young or alternative ones or both.

Alternatively, neurofunctional reorganization phenomena are attributed to reduced neural efficiency, also known as dedifferentiation, resulting in reduced performance in the old [31, 32, 34, 83, 84]. According to the dedifferentiation hypothesis, aging reduces the specialization of neurons which is critical for their optimal functioning [31]. Accordingly, increased activations could be the result of randomly recruiting neurons in an attempt to meet processing demands [19], or could reflect the brain's failure to selectively recruit specific regions [34] whereas increasing task demands may aggravate the non-specificity of neural responses [85]. Evidence exists to support the idea that neural responses are less specific in older adults when compared with younger ones, as demonstrated in the ventral visual cortex during a viewing of pictures task [83, 86], during a working memory task [87] (for a review, see [88] and in motor evoked potentials [89]). It is not clear however whether this loss of neural specificity would be the result of aging or could be attributed to larger experience of older adults in recognizing objects [83]. At the same time, it is thought that both compensation and dedifferentiation phenomena may take place in the same person simultaneously in different regions [87]. The dedifferentiation account would predict a reduction in performance together with an increase in activation, thus resembling the descending part of the inverted-U shape relation between task demands and fMRI activation, as per CRUNCH.

An additional explanation for age-related functional reorganisation is that aging selectively affects the default mode network (DMN). This network is normally activated during a situation when one is not involved in any task but instead monitors their internal and external environment [7] and deactivated when performing cognitive tasks so as to reallocate attentional resources towards them [35]. It is thought that the semantic network is largely activated at rest, as individuals would be engaged in language-supported thinking when not performing specific

tasks [90]. It has been found that when the task is cognitively demanding, DMN deactivations are smaller and slower for older adults, implying that they are more easily distracted whereas their capacity to inhibit irrelevant information is compromised [28, 35, 91], in line with the inhibitory control view [92] and the cognitive theory of aging [7]. In difficult semantic tasks, maintained performance was associated with increased segregation between DMN and semantic control regions at rest, whereas reduced performance was associated with increased verbal thinking at rest [93]. It is possible that aging reduces the efficiency of transferring attention away from resting areas towards task requirements, thus probably affecting the balance between DMN and task-related activity and resulting in reduced cognitive performance [7].

The neurofunctional reorganization proposals discussed above seem to be exclusive of another as they tend to focus and attribute meaningfulness in increased or decreased activation in isolated brain regions, whereas none seems to fully capture and explain age-related reorganization [94]. Several researchers have attempted to identify the 'common factor' [95] in age-related brain activation patterns to explain reorganization. Cabeza (2002) [19] considers that functional reorganization is more likely to be non-intentional and neuron-originated rather than a planned change of cognitive strategies, since it is manifested in simple tasks or following unilateral brain damage, over which one has little control. On the contrary, Reuter-Lorenz and Cappell (2008) [28] consider unlikely that such a huge variability in brain activation stems from the same 'common factor' or is due to age-related structural changes in the brain, because then it would be consistent across all tasks. Instead, aging seems to selectively affect specific regions, mainly default-mode regions and the dorsolateral PFC [7] whereas inter-individual variabilities need to be emphasized when accounting for age-affected cognitive domains [96].

Recent studies tend to combine data on functional, structural and lifetime environmental factors to explain reorganization in a more integrative manner. In this direction, the more comprehensive Scaffolding Theory on Aging and Cognition- STAC hypothesis proposes that aging is no longer characterized by uncontrollable decline of cognitive abilities because the brain develops its own resilience, repairs its deficiencies and protects its functions [28, 97]. This idea is reflected in the aging models that emphasize the plasticity of the brain due among other factors to training interventions and their impact on neural structure, as well as functional and behavioral outcomes [98–100]. The impact of short-term practice as well as long-life training would impact young and older adults differently [69]. Accordingly, engaging in intellectually challenging activities and new learnings throughout the course of a lifetime but also on a shorter-term course could stimulate plasticity of the brain. The capacity of the brain to resolve the mismatch between intellectual demands and available neurofunctional resources and its capacity to trigger behavioral adaptive strategies, would define its plasticity and affect its brain knowledge systems and processing efficiency [69]. Plasticity would demonstrate itself as increased functional activation especially in regions that are most structurally affected by aging because of atrophy, loss of grey and white matter density and cortical thinning, such as in the fronto-parietal network [99]. Aging could thus be characterized by structural loss but also neural and functional adaptation to this loss, including through the utilization of new strategies [99]. Indeed, age-related overactivations seem to be a reliable and consistent pattern observed in multiple domains regardless of whether they are more localized, contralateral or seen in the fronto-parietal multiple-demand network [101]. In summary, the more adaptable and the more dynamic the brain is, the better it would maintain its cognitive abilities [102].

Specifically to semantic memory preservation in aging, it is not clear what mechanisms are in place to account for the preservation of semantic memory in aging, supported by the intersection of both domain-general and linguistic abilities [66]. Findings in the literature about the adoption of neurofunctional activation pattern during semantic processing in aging, vary. Two additional compensatory hypotheses have been proposed: the executive hypothesis refers

to the recruitment of domain-general executive processes seen as overactivation in prefrontal, inferior frontal and inferior parietal brain regions to compensate for age-related cognitive decline [6, 103], as seen for example in a semantic judgment task [56]. Indeed, the metaanalysis of semantic memory studies performing activation likelihood estimation (ALE) between young and older participants [22], found a shift in activation from semantic-specific regions to more domain-general ones, in line with the executive hypothesis. The semantic hypothesis on the other hand, also known as left anterior-posterior aging effect (LAPA), refers to the recruitment of additional semantic processes in older adults, seen as overactivation in 'language' regions in the left posterior temporo-parietal cortex [104, 105]. Given the larger decline in older adults of executive over language functions could justify this latter hypothesis considering that language is better maintained over executive processes [106]. Evidence for the semantic hypothesis was found in a study using semantic judgment task where participants had to decide if a word is an animal or not. Older participants had more bilateral parietal, temporal and left fusiform activations than younger ones who presented more dorsolateral activations, which the authors interpreted as older participants relying more on semantic processes whereas younger ones relying more on executive strategies [107]. However, language and executive functions are overall intertwined given that regions such as the left inferior frontal gyrus and the PFC are proposed to serve both executive and language functions, thus blurring the intersection between the semantic and executive hypothesis [53].

An alternative approach can be seen within the good-enough theory, which claims that participants tend to construct semantic representations which are 'good-enough' or shallow rather than more complete or detailed ones, with the aim to perform the task at hand with the least effort and save on processing resources [108–110]. This theory refers to overall language processing, but it could also be applied to the semantic representation of words as inferred by the semantic judgment task used for the current study. Accordingly, participants and especially older adults at increased task demands, may resort to a more 'shallow' or superficial interpretation of the semantic judgment task they are required to perform and instead of analyzing thoroughly all semantic aspects of the words they are presented with (e.g. semantic features of the apple in comparison with the grape or cherry), may bypass some aspects of the task and thus resort to a quick decision. Such a shallow processing could be manifested with decreased activation overall, as well as in the semantic control network which would be in charge of selectively controlling for semantic features while ignoring others [56]. This alternative explanation is in line with the idea that at peak levels of demand, participants may become frustrated with frequent errors or difficulty to resolve competing representations, and may deploy inefficient strategies [111].

In summary, some inconsistencies are found in interpretation of results, with both increased and decreased activation reported as the result of aging [7, 112]. Neurofunctional reorganization can take the form of both inter- and intra-hemispheric changes in activation and manifests as both increased and decreased activation of specific regions [7]. When performance is compromised, reduced activation is interpreted as impairment, attributed to neural decline, inefficient inhibitory control or de-differentiation [28] whereas when performance is maintained, it is claimed to be compensatory. Most studies seem to agree on increased activation, interpreting it as compensatory and positive, whether it is understood as increased attention or as suppression of distracting elements [113]. Overactivation is also found in Alzheimer's disease (AD) and mild cognitive impairment (MCI) patients demonstrating either its compensatory role or a progressive pathology predicting further decline [34, 35]. It seems that neurofunctional reorganisation of the aging brain is more complex and further research is still required to be able to 'draw' a pattern of activation that integrates the existing findings in a

comprehensive model and one that can be applied to semantic memory, one of the best preserved cognitive fields in aging.

## Current study

The aim of this study is to identify whether aging affects the brain activity subserving semantic memory in accordance with the CRUNCH predictions, through a semantic judgment task with two levels of demands (low and high). Task demands will be manipulated through semantic distance, which is found to influence both performance and brain activation levels [49, 61, 67, 114–117]. We hypothesize that brain activity and behavioral performance (dependent variables) will support the CRUNCH model predictions when demands on semantic memory are manipulated in young and old adults (age and task demands: independent variables). More specifically, it is expected that 1) the effects of semantic distance (low vs. high-demand relations) on neurofunctional activation and behavioral performance (accuracy and RTs) during the semantic judgment task will be significantly different between younger and older participant groups, with younger adults performing with higher accuracy and faster response times than older adults. Furthermore, we predict age group differences in brain activation in semantic control regions bilaterally which are sensitive to increasing task demands [24]. This will be evident with a significant interaction effect between age group and task demands within regions of interest consisting of the core semantic control regions: IFG, pMTG, pITG and dmPFC. This will support the idea of the brain's declining ability to respond to increasing task demands with advancing age. If this interaction is not found between task demands and age, the following are expected 2) In the low-demand (LOW) condition, both younger and older participants will perform equally in terms of accuracy and with less errors than in the high-demand condition. However, it is anticipated that older adults will present longer RTs and significant increases in activation in left-lateralized semantic control regions compared to the younger participants. 3) In the high-demand (HIGH) condition, it is expected that younger adults will perform better (higher accuracy and lower RTs) and present significant activation in the left-hemisphere semantic control regions compared to older adults. Older adults are expected to exhibit reduced performance compared to younger adults (lower accuracy and higher RTs), reduced activation in left-lateralized semantic control regions, and increased activation in right-lateralized semantic control regions compared to the younger adults. To illustrate the hypothesized relations between task demands and accuracy, RTs and activation in young and older adults, see Figs 1–3 below. The theoretical relations between task demands and activation are represented in the decrease in activation in the left hemisphere (cross-over interaction, Fig 3) and the increase in activation in the right hemisphere (difference in slopes interaction, Fig 4), confirming the hypothesized CRUNCH predictions. These portray the main effects of age and task demands as well as their interaction highlighted by thick lines.

These analyses are looking for age and load effects on task performance and on brain activation in separate analyses. Follow-up exploratory analyses within the ROIs will explicitly test how differential brain activation is related to task performance. It is hypothesized that older adults who have high levels of brain activation in left-lateralized semantic control regions during the high-demand condition, similar to the young adults, will have higher levels of task performance (reduced errors and RTs) than their counterparts whose brain activation is lower in these regions, as per the CRUNCH model, indicating that they have not yet reached their crunch point after which performance and activation decline. To accept the above hypotheses, at least one ROI from the ones mentioned is expected to be activated.

A control condition is part of the task and was designed to maximize perceptual processing requirements and minimize semantic processing ones [118, 119]. As a test of positive control,

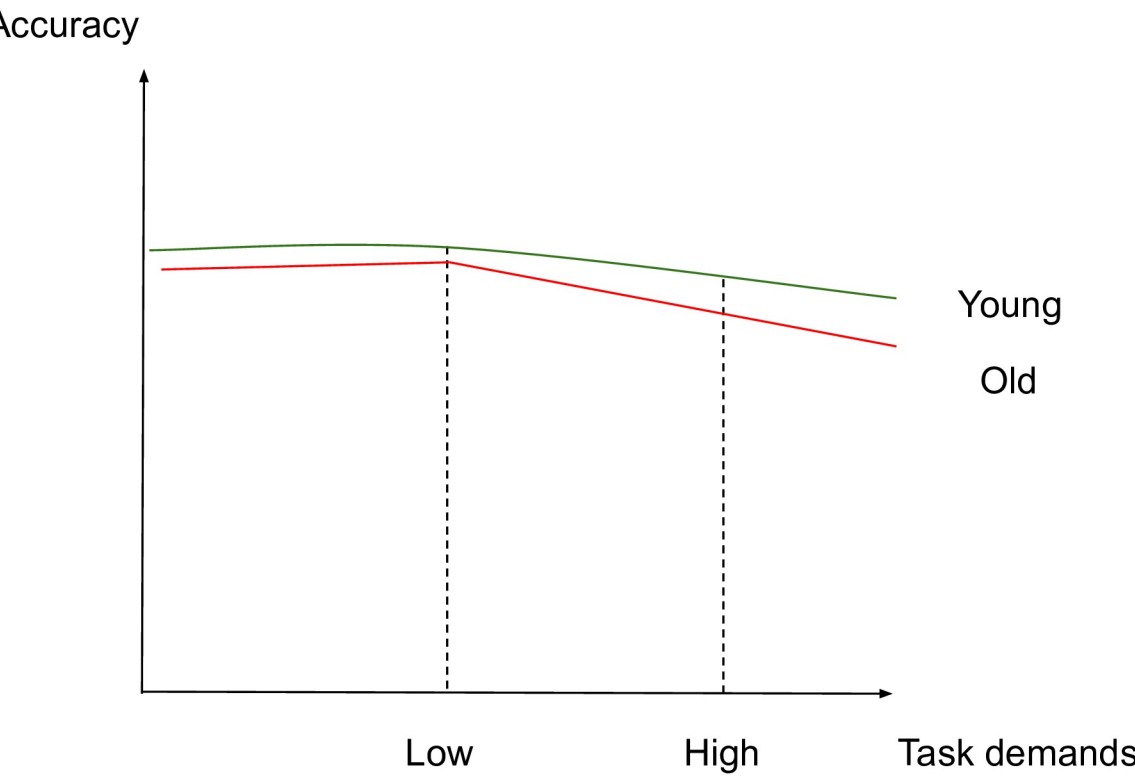

**Fig 1. Accuracy and task demands in younger and older adults.**

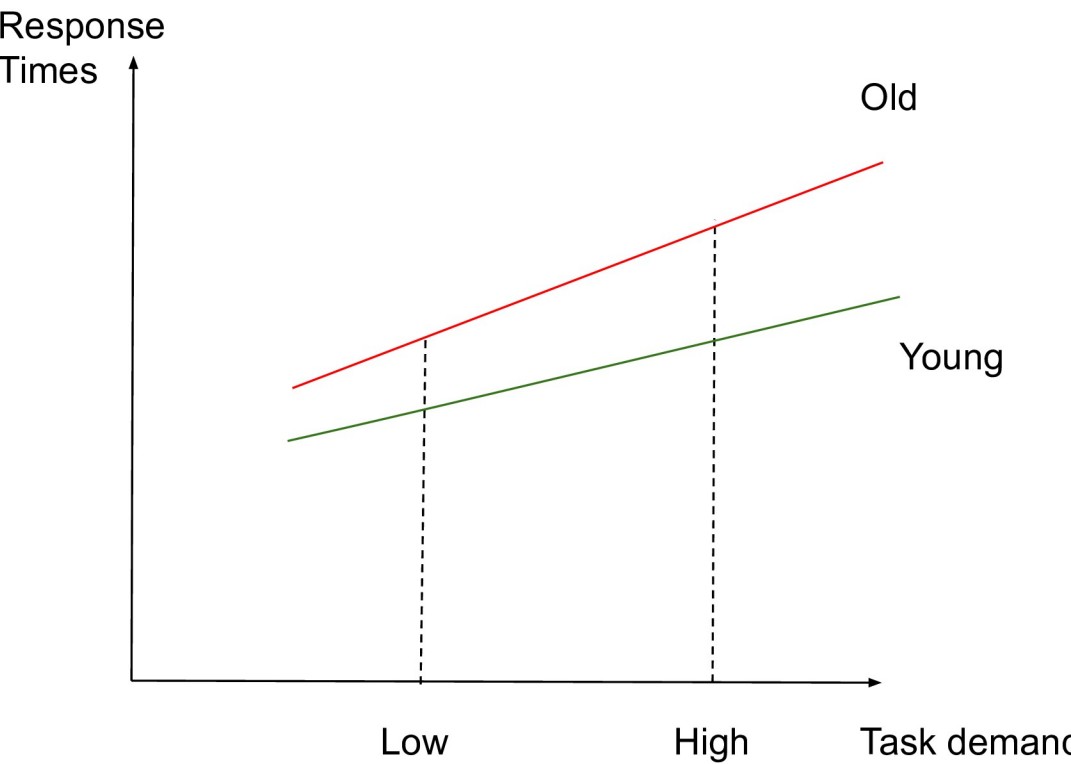

**Fig 2. RTs and task demands in younger and older adults.**

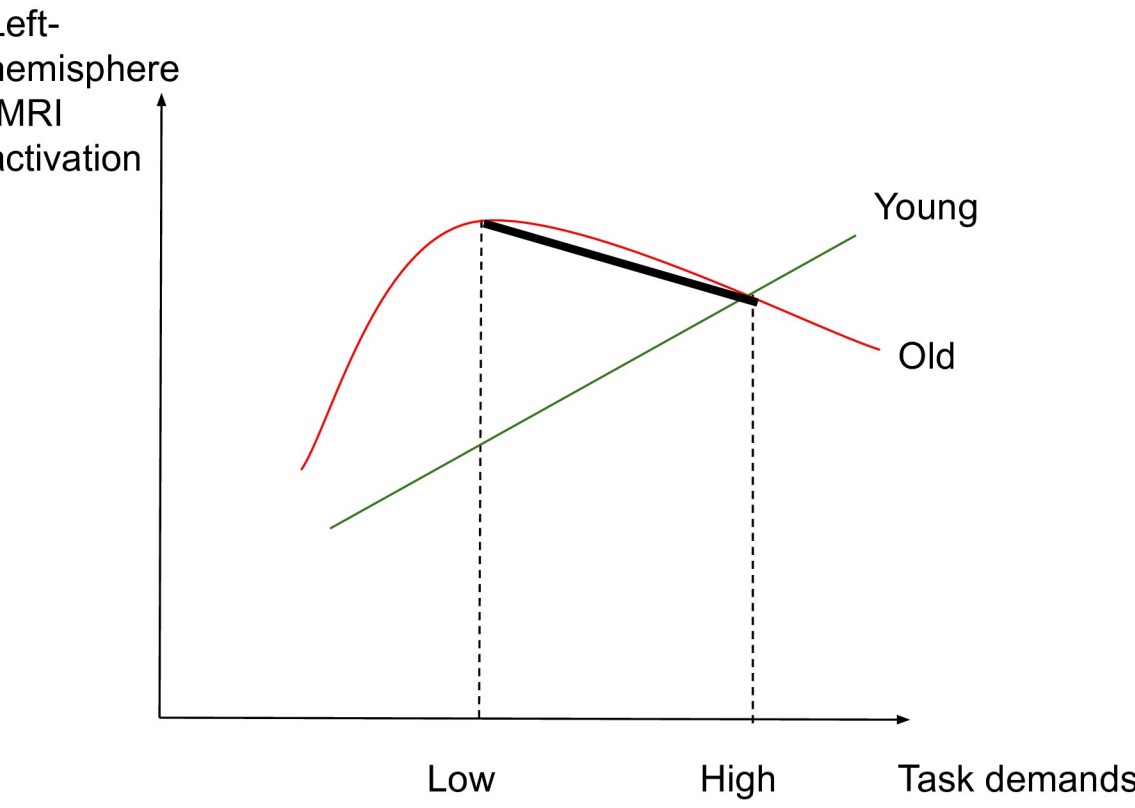

**Fig 3. Left-hemisphere activation and task demands in younger and older adults.**

within group comparisons with the control condition are expected to show activation in the primary visual and motor cortices, which are involved with viewing of the stimuli, response preparedness and motor responses [64, 120, 121]. No CRUNCH effects are expected in the control condition. Task effects within each age group will also be tested and activation is expected to be of greater amplitude in the high vs. low condition in both young and old age groups.

This task design utilizes explicit definitions of low and high levels of task demand. However, each individual participant will experience their own subjective level of task difficulty. Perceived difficulty of triads will be measured on a difficulty 1–7 likert-scale (e.g. 1: very easy, 7: very difficult. Subsequent analyses will explore this question with heterogeneous slopes models using individualized rescaled levels of task difficulty and will compare brain activation with performance, brain activation with perceived difficulty and performance with perceived difficulty. This approach will determine how the relationship between individual task difficulty and brain activity is affected by age group.

## Proposed experiment: Materials and methods

The authors comply with the Centre de Recherche Institut Universitaire de Gériatrie de Montréal (CRIUGM) Ethics Committee and the Centre intégré universitaire de santé et de services sociaux du Centre-Sud-de-l'Île-de-Montréal requirements (CÉR-VN: Comité d'Éthique de la Recherche- Vieillissement et Neuroimagerie), in line with the principles expressed in the Declaration of Helsinki. The ethics committee of CRIUGM and CÉR-VN approved this study with number CER VN 16-17-09. The approval letter is available in the

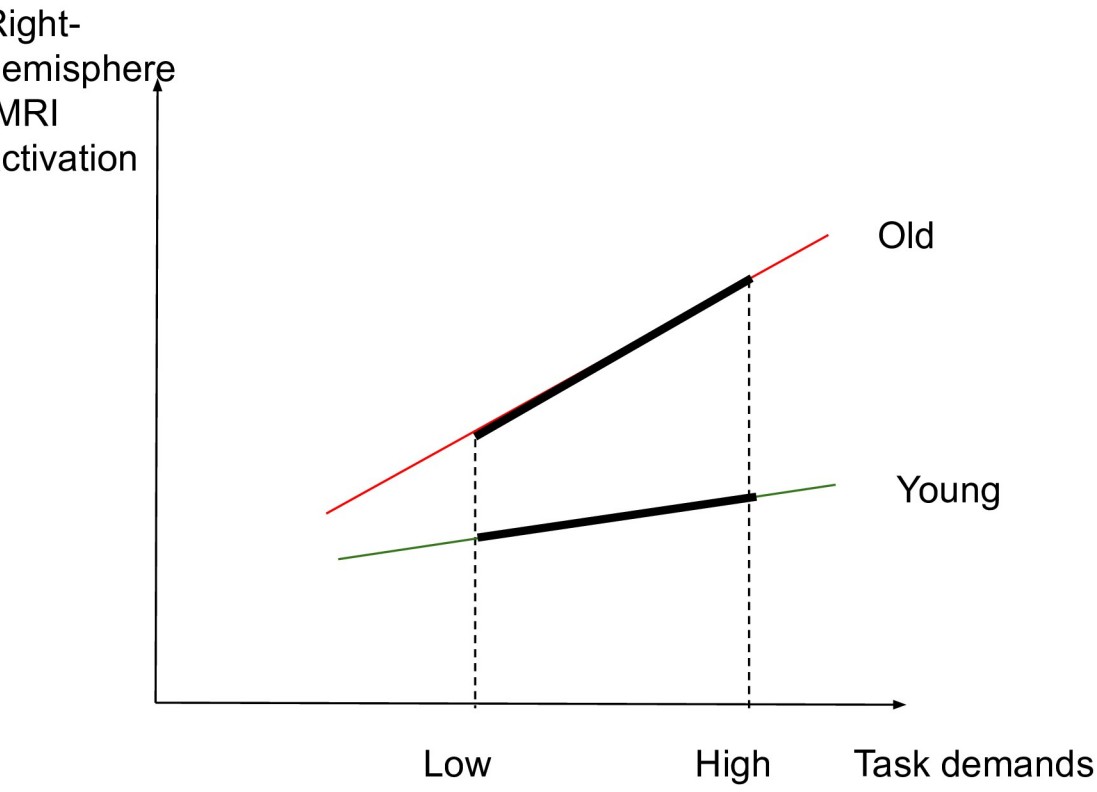

**Fig 4. Right-hemisphere and task demands in younger and older adults.**

OSF repository (DOI: 10.17605/OSF.IO/F2XW9). For all methodology aspects of this current study, compliance with the OHBM COBIDAS report/checklist [122] and guidelines [123] will be aimed for as much as possible. We will share the preprocessed functional data-sets in MNI space publicly in Open Science Framework (https://osf.io/) with a digital object identifier (DOI: 10.17605/OSF.IO/F2XW9) to permanently identify the dataset [122], and we will index it at the Canadian Open Neuroscience Platform (https://conp.ca/) to increase findability. In addition, once these become available, we will upload our unthresholded sta-tistical maps to neurovault (https://neurovault.org/), an online platform sharing activation data. Permanent links to the unthresholded statistical maps to be uploaded at Neurovault will be provided as part of the dataset deposited on the OSF, under the same DOI (DOI: 10.17605/OSF.IO/F2XW9). Data will be organized following the Brain Imaging Data Structure (BIDS) to maximize shareability. Supporting documentation for this study is available at DOI: 10.17605/OSF.IO/F2XW9.

## Participants

A sample of 80 participants will be tested for this study: 40 in each group, Young: 20–35 years old and Older: 60–75 years old (male = female). We will recruit 86 participants assuming that some will be excluded in the process due to low task performance, excessive motion or technical issues. Participants will be recruited through the Centre de Recherche Institut Universitaire de Gériatrie de Montréal (CRIUGM) 'Banque de Participants', but also through poster announce-ments posted in Montreal and in social media. Participants will be bilingual (French and English-speaking) with French as their dominant language used on a daily basis. Multilingual

participants will be excluded, as speaking many languages may influence semantic performance [124]. Participants will be matched for education level with college studies (CEGEP) as a minimum level of education, since education is a measure of cognitive reserve [82]. Participants will undergo a series of neuropsychological and health tests to determine their eligibility for the study as inclusion/exclusion criteria:

- A health questionnaire (pre-screening to take place on the phone) to exclude participants with a history of dementia, drug addiction, major depression, stroke, aphasia, cardiovascular disease, diabetes, arterial hypertension or any drugs that could affect results. The pre-screening includes questions for bilingualism and use of French language, which needs to be the dominant one (inclusion criteria) (the complete questionnaire is available on osf.io, DOI: 10.17605/OSF.IO/F2XW9).

- The Edinburgh Handedness Inventory scale: participants will be right-handed with minimum score for right-handedness of 80 [125].

- The MoCA (Montreal Cognitive Assessment) test with a minimum cutoff score of 26 [126, 127].

- The MRI-compatibility checklist (Unité Neuroimagerie Fonctionnelle/UNF) test (available at https://unf-montreal.ca/forms-documents/).

    The following tests will also be performed with participants:

- The Similarities (Similitudes) part of the Weschler Adult Intelligence Scale (WAIS-III) test [128, 129]

- The Pyramids and Palm Trees Test (PPTT) (version images) [130] will be used as a measure of semantic performance.

- The questionnaire Habitudes de Lecture (Reading Habits) (based on [131] as a measure of cognitive reserve [82].

    Participants will provide written informed consent and will be financially compensated for their participation according to the CRIUGM and Ethics Committee policies.

## Power analysis

This sample size is based on power calculation results from an age group comparison on a similar semantic task [132]. This dataset used a Boston naming semantic task and compared healthy young and old age groups. From this dataset effect size estimates were calculated from the contrasts for high versus low task demands within and between age groups. Effect sizes were extracted from the primary regions of interest for this study as defined by a recent meta-analysis of semantic control [24]. From the identified locations, a 10 mm cube was defined to identify the effect size at the published location, mean effect size and the robust maximum effect size in the ROI. Statistical power was then estimated using the G*Power tool [133]. Within group measures had robust effect sizes and demonstrated that sufficient power (alpha = 0.05, beta = 0.90) was achieved with a sample size of 40 participants in each group. The between group comparison of differential activation had sufficient power within bilateral temporal gyri and medial PFC. In addition, the proposed study will use more than twice the number of trials used in the data used for power estimations. This will decrease the within participant variation and will increase the power above that provided by the [132] study. The table of effect sizes used for the power analyses for within and between group comparisons are included as supplementary material at OSF.

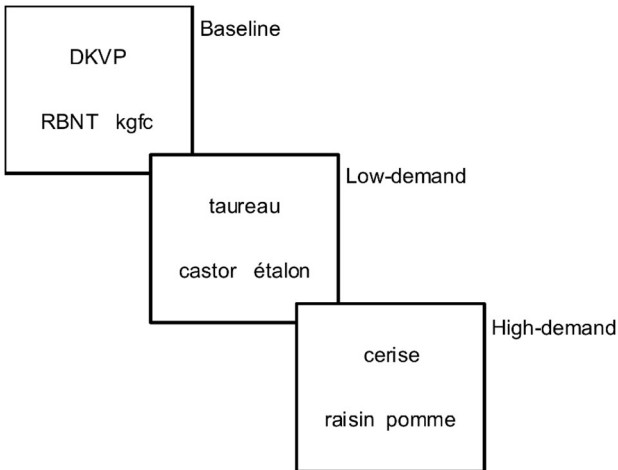

**Fig 5. Examples of triads.**

## Materials

Participants will be administered a task of semantic similarity judgment in French and that is suitable for the Quebec context developed for the current study, similar to the Pyramids and Palm Trees test (PPTT) [130]. The task proposed here involves triads of words resembling a pyramid where participants will need to judge within a time limit of 4 seconds which of the two words below (target or distractor) is more related to the word above (stimulus). Both target and distractor words are associated in a semantic relation with the stimulus word. Participants will thus be required to select which of the two competing words has a stronger semantic relationship to the stimulus word as measured by semantic distance between the stimulus and the distractor. Two types of triads exist: a) low-demand (distant) relations: the more distant the semantic relation between stimulus and distractor, the less demanding will be to select the correct target and b) high-demand (close) relations: the closer the semantic relation between stimulus and distractor, the more demanding will be to select the correct target as competition between the target and distractor words will be higher [61].

The task (150 triads in total) has two experimental conditions (120 triads: 60 low-demand (LOW) and 60 high-demand (HIGH) semantic relations) and one control condition (30 triads). For the control condition, the task will be to indicate which of the two consonant strings, which will be presented pseudo-randomly, are in the same case as the target strings (e.g. DKVP: RBNT-kgfc). The stimuli will look like Fig 5 below:

**Stimuli description.**   The stimuli were developed for the current study. In every condition, the targets and distractors were matched for: a) Type of semantic relation: taxonomic and thematic. For thematic relations, the semantic distance was calculated with the help of a dictionary: 'Dictionnaire des associations verbales (sémantiques) du français' (http://dictaverf.nsu.ru/dict, version accessed on 2014), as a function of the number of respondents that associated two words together (i.e., the larger the number of respondents, the more closely associated the two words are, and vice versa). As such, a score of 1 means that only one person provided this answer (distant thematic relation) whereas a score of 100 means that 100 people provided this answer (close thematic relation). b) Frequency, based on the Lexique 3 database referring to films [134] c) Imageability, based on the Desrochers 3600 database [135]. Additional imageability ratings were collected from 30 participants for items without ratings in the above database. A Pearson's correlation was performed with 30 test words from the Desrochers database

to confirm that the ratings given for the new words were relevant compared to the ones that already exist. Participants rating items with a correlation value less than 0.6 were excluded, as it was deemed that they were not concentrated on the task. The final imageability rating of an item was the mean of the scores given by all included participants. ANOVA and Bonferroni corrected Tukey tests were performed to ensure the matching of a target and distractor for every condition. Finally, targets and distractors were matched on d) Word length.

The stimuli were created in a gradual process, continuously testing and evaluating its adequacy and aiming for a less than 40% error rate with pilots to test it, measure response times and gather comments. Every time, the four conditions were matched and passed an ANOVA test for mean frequency, imageability and length. Also, pilot participants were asked questions about the duration of the task and the sufficiency of time to respond. To evaluate the validity of the stimuli pertaining to low vs. high demands and younger vs. older adults, a pilot evaluation of stimuli was conducted by 28 participants (14 were older adults, age range: 67–79 years old, female = 9 and 14 were younger, age range: 21–35 years old, female = 10) for 60 triads (30 low-demand and 30 high-demand) using E-Prime. Repeated measures analysis of variance (ANOVA) was applied to the mean accuracy and median response data within each level of task demand (control, low, high) across the two age groups. The results are described below:

*Accuracy*. The Greenhouse-Geisser estimate for the departure from sphericity was $\varepsilon = 0.63$. There was not a significant interaction between age group and task demand, $F(1.27, 32.94) = 0.065$. $p = 0.85$, $\eta2 = 0.0025$. The main effect of task demand was significant, $F(1.27, 32.94) = 10.36$, $p = 0.0015$, $\eta2 = 0.28$. The estimated marginal means were: Control = 0.84, Low = 0.80 and High = 0.72. The main effect of age group was not significant, $F(1, 26) = 0.34$, $p = 0.57$, $\eta2 = 0.013$.

*Response times*. The Greenhouse-Geisser estimate for the departure from sphericity was $\varepsilon = 0.54$. There was not a significant interaction between age group and task demand, $F(1.08, 28.14) = 1.14$. $p = 0.30$, $\eta2 = 0.042$. The main effect of task demand was significant, $F(1.08, 28.14) = 49.38$, $p < 0.0001$, $\eta2 = 0.66$. The estimated marginal means were: Control = 1390ms, Low = 2230ms and High = 2292ms. The main effect of age group was significant, $F(1, 26) = 4.78$, $p = 0.038$, $\eta2 = 0.15$.

Based on the above pilot data, we confirm that our task includes a load effect that impacts task performance (accuracy and RTs) differently between younger and older adults, in the expected directions.

The following definitions were used:

Low-demand (distant) triads:

- For taxonomic relations:

All items (stimulus, target, distractor) belong in the same semantic category (e.g., animals). Stimulus and target words belong in the same semantic sub-category (e.g. birds). For example, taureau: ÉTALON-castor (bull: STALLION-beaver).

- For thematic relations:

Both the target and distractor words are thematically related to the stimulus and belong in the list of answers referred by dictaverf. To ensure the biggest distance possible, the target was the first adequate answer mentioned in dictaverf, whereas the distractor was the last or closest to the last answer, meaning that it had a score close to 1. For example, sorcier: village-BAGUETTE (wizard: village-WAND).

Alternatively, to ensure the biggest distance possible, the following criteria were used: when the distractor word is 1 (which means only 1 person provided this answer), when the distractor

word is between 2–5 and the target word is above 10, and when the difference between the target and distractor words is bigger than 100.

High-demand (close) triads:

- For taxonomic relations:

All items in the triad come from the same semantic sub-category (e.g. birds). The stimulus and target items share a visual or structural feature whereas the distractor word does not. For example, 'cerise: RAISIN-pomme' (cherry: GRAPE-apple) where cherries and grapes have a similar size and bunch structure.

- For thematic relations:

Both the target and distractor words are thematically related to the stimulus. The target was the first adequate answer mentioned in dictaverf whereas the distractor had a score smaller or equal to half of the score of the target and was bigger or equal to 4. This criterion was used to ensure that the distractor was a more frequently mentioned answer but distant enough from the target (e.g. half of the people mentioned the distractor as opposed to mentioning the target). For example, 'enfant: JOUET-sourire' (child: TOY-smile).

## Experimental design

**Session 1: Neuropsychological tests.** Participants will be recruited through the CRIUGM pool of participants and public announcements, with initial eligibility assessed through a phone interview (health questionnaire and MRI compatibility form). If eligible, the participant will partake in the first experimental session (approximately 90 minutes), during which they will sign the informed consent and MRI-compatibility forms, complete neuropsychological tests (see Participants section above) and practice with 15 practice triads (5 for every condition). Participants who qualify (meet the inclusion criteria from health questionnaire, MRI-compatibility questionnaire, MOCA and Edinburgh Handedness Inventory scale) for the fMRI scanning session following tests will proceed with the second session one week later (maximum 2 weeks later).

**Session 2: fMRI scanning.** For the second experimental session, the time commitment from the participant: is 90 minutes to allow for practice with triads, getting ready and leaving, following COVID-19 requirements. During this session, participants will listen to task instructions, and practice with 3 triads (1 per condition). Participants' vision will be corrected, if necessary, with MRI-175 compatible lenses according to their prescription shared from the previous session. Additionally, pregnancy tests will be carried out when relevant, earplugs will be given to reduce machine noise and instructions will be given to remain still in the scanner while foam rubber pads in the head coil will restrict movement. Participants will then proceed with the actual task in the scanner. Stimuli will be presented with E-Prime 2.0.10.356 software run on Microsoft Windows 10 through an LCD projector projecting to a mirror over the participant's head. Participants will select their responses using the index fingers of both hands on the MRI-compatible response box. A response on the right will be with their right hand and a response on the left with their left hand. Response data and response times (RTs) will be recorded via E-Prime for further analysis. No feedback will be shared with participants. Participant testing will alternate between young and older adults to minimize any bias due to scanner changes/upgrades.

The semantic task will be event-related. Triads will be presented for 4 seconds, during which participants will need to make their choice by pressing on the left or the right button to select the word on the left or right respectively. A black screen will follow for approximately

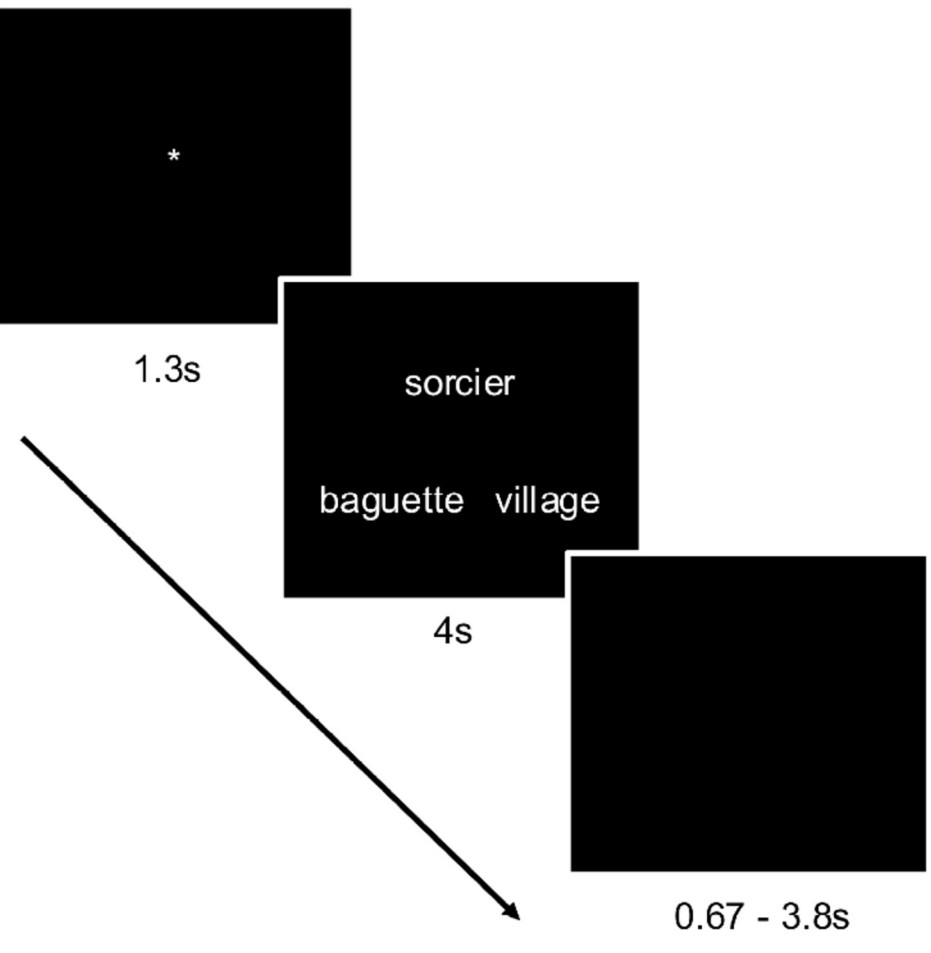

**Fig 6. Example of trial.**

2.2s (this interstimulus interval (ISI) will vary randomly between 0.67s and 3.8s to minimize possible correlations with the BOLD signal). A fixation point will appear for 1.3s to prepare the participant for the next trial. The whole trial will last between 5.97s and 9.10s, with an average of 7.5s. See below for a description of the methods used to determine the ISIs. Black screens were included at the beginning and the end of the runs. Information on the scanning flow is available in Fig 6 below:

The task will be split in two runs with 75 triads per run (30 low-demand (LOW), 30 high-demand (HIGH) and 15 control triads), interleaved in a pseudo-random fashion. The duration for every run will be 9:45 minutes. The whole session is expected to last 45 minutes, including a 5-min break between runs 1 and 2.

**Session 3.** In regards to perceived task difficulty, an additional session with participants one week following the fMRI acquisitions will take place, whereby they will rate each triad on a difficulty 1–7 likert scale (eg. 1: very easy, 7: very difficult). We will further assess whether perceived difficulty correlates with actual performance scores (accuracy rates and RTs) and whether perceived difficulty correlates with levels of activation in the young and older adults (e.g. whether increased levels of perceived difficulty correlate with increased RTs and reduced accuracy, as well as levels of activation in semantic control regions).

**Stimuli order and ISIs.** To design the experiment in a way that maximizes design efficiency, optimal trial ordering and interstimulus intervals (ISIs) were chosen [136]. The

methodology used simulated designs of random ordering of the three conditions. In addition, the ISIs were randomly drawn from Gamma distributions across a range of parameter values (shape: 0.1 to 10, scale: 0.1 to 5). This approach included expected error rates produced during the stimuli pilots to maximize design efficiency in the face of errors. A total of 800,000 simulations were performed. The ISI distribution and specific list as well as the condition order in which there was the smallest decrease in required BOLD signal response for detection as errors increased were chosen. The related ISIs are uploaded to the OSF platform.

**fMRI data acquisition.**   Functional scans will be performed on a 3Tesla Syngo MR E11 Prisma_fit Siemens MRI machine with 32 channels at UNF (Unité de Neuroimagérie Fonctionnelle), CRIUGM. The start of the stimulus presentation software will be triggered by a pulse sent from the MRI to the stimulus laptop. To detect effects between conditions and to ensure a good fMRI signal in the brain, pilot data collected using the scanning protocol described here suggested a minimum temporal signal to noise ratio (TSNR) of 20 throughout the brain [137]. Participant data will be excluded if TSNR, assessed from every participant's time series, is below 20. We will acquire T1-weighted MRI images for co-registration with fMRI data and atlases and to identify ROIs to be used as masks in the functional data analysis. An meMPRAGE (multi-echo MPRAGE) sequence (704 total MRI files) will be acquired with 1x1x1mm resolution, 2.2s repetition time, 256x256 acquisition matrix, a Field of View (FOV) of 256mm covering the whole head and echo times of 1.87ms, 4.11ms, 6.35ms, 8.59ms, 13ms and 15ms. The phase encoding orientation will be sagittal with a flip angle of 8 degrees.

For the functional scans (run 1 and 2), T2-weighted BOLD data will be acquired on the entire brain (including the cerebellum) using an Echo Planar Imaging (EPI) sequence with 50 slices, resolution 2.5x2.5x3mm, echo time of 20ms, repetition time of 3s and a flip angle of 90 degrees. Field of view will be 220x220mm and the acquisition matrix will be 88x88, in AC-PC direction covering 150mm in the z-direction. Slice order will be ascending-interleaved. For each run, 195 scans will be collected. The SIEMENS default double-echo FLASH sequence for field map distortion correction with the same parameters will be acquired after each sequence for inhomogeneity correction. Functional images will be reconstructed to the collected matrix size with no prospective motion correction. Two initial dummy scans will be collected and discarded by the MRI allowing for T1 saturation.

## Proposed analyses

### Behavioral data analysis

Response times and accuracy rates will be collected for every participant. Sex will be used as a covariate in all analyses. To account for performance, brain imaging analysis will focus on correct trials only ensuring that we are looking at brain activation related to accurate performance. Behavioral data (RT and accuracy) will be analyzed using mixed- design ANOVA with age as a between-subjects factor and condition (high vs. low demands) as within-subject factor. Accuracy rates will be transformed using Fisher logit approximation to avoid ceiling effects. Group analyses of the imaging data will be performed including behavioral covariates to investigate age group differences in the relationships between brain activity and task performance. Multiple comparisons across the 40 ROIs will be made using false discovery rate adjustments. Analyses will explicitly focus on the relationships between brain activation and task performance. These analyses will identify brain regions where age group differences in activation are dependent or independent of task performance. Time-outs (delayed responses) will be modeled and analyzed separately. Any missing or incomplete data will be excluded (the whole participant).

### Imaging data analysis

**Preprocessing.** Preprocessing image analysis will be performed with SPM12 software. Images will be corrected for slice timing (differences in slice acquisition time), with ascending-interleaved slice order and using the acquisition time for the middle slice as the reference. We will use field map correction to correct EPI images for distortion using the Calculate VDM toolbox and the first EPI image as reference. The gradient field map images will be pre-subtracted by the scanner to provide phase and magnitude data separately. Motion correction will be applied for within-subject registration and unwarping. Motion parameters will be used later as confound variables. Data will be visually inspected for excessive motion. Participants with estimated acute motion parameters of more than 2mm, or 1-degree rotation, between scans in any direction, will be excluded. EPI functional volumes will be registered to the average anatomical volume calculated by the machine over the 4 echoes of meMPRAGE T1-weighted anatomical scan. The mean anatomical image will be used as the reference image and as quality control. Anatomical variations between subjects will be reduced by aligning the structural images to the standard space MNI template, followed by visual inspection of their overlay. An 8mm full width at half maximum (FWHM) Gaussian blur will be then applied to smooth images within each run. The final voxel size after preprocessing will be 3x3x3 mm.

**fMRI data analysis.** fMRI data analysis will be performed with SPM12 focusing on semantic control primary ROIs. Using files created by E-Prime during stimulus presentation, stimulus onset files will be created and regressors will be defined. For the 1st level (intrasubject) analysis, a General Linear Model (GLM) employing the canonical Hemodynamic Response Function (HRF) and its derivative both convolved with a model of the trials will be used to estimate BOLD activation for every subject as a function of condition for the fMRI task. The inclusion of the derivative term accounts for inter-individual variations in the shape of the hemodynamic response. Correct trials will be modeled separately for low demand and high demand conditions. Incorrect trials for low and high demands will be modelled together in their own regressor and not investigated further. Each participant's fMRI time series (2 runs) will be analyzed in separate design matrices using a voxel-wise GLM (first-level models). Movement parameters obtained during preprocessing, and their first and second derivatives, will be included as covariates (regressors) of no interest to reduce the residual variance and the probability of movement-related artifacts. A high-pass filter with a temporal cut-off of 200s and a first-order autoregressive function correcting for serial autocorrelations will be applied to the data before assessing the models. Two contrasts of interest will be calculated collapsing across the two runs. These contrasts will be low-demand, correct trials > control and high-demand, correct trials > control. These contrasts will be used for second level group analyses to compare age group and effects of task demand.

The analysis will first test for an interaction between age group and task demands. A significant finding will support hypothesis one. It is expected that a significant interaction will be driven by significant post-hoc t-tests of age group within the low-demand condition, where the older age group will have significantly greater activation than the younger age group in left semantic control regions. This finding will support hypothesis two. It is also expected that there will be a significant post-hoc t-test of age group within the high demand condition where the young age group will have significantly greater activation than the old in the left semantic control regions. It is also expected that the old age group will have significant greater activation in the right semantic control regions. This finding will support hypothesis three.

To account for differences in HDR between younger and older adults, the event-related first-level statistical model of the fMRI data will include the event-chain convolved with the double-Gamma hemodynamic response function and its first derivative. The inclusion of this

extra regressor will capture variance in the data due to any inter-participant or inter-group variations in the shape of the hemodynamic responses.

**Defining the anatomical/functional ROIs.** This study's hypotheses depend on ROIs that include semantic control regions associated with low and high-demand conditions. To identify ROIs of the semantic control network demonstrating demand related differences in brain activation, the results of a recent meta-analysis will be used [24]. This analysis utilized data from 126 comparisons and 925 activation peaks and is the most comprehensive and up to date meta-analysis of semantic control networks. The results identified twenty highly significant peak locations throughout the inferior frontal gyrus, insula, orbitofrontal cortex, precentral gyrus, middle and inferior temporal gyri and the fusiform gyrus, see Table 1 [24] for specific x, y,z locations. Spheres of diameter of 10mm will be created at each of these locations and the corresponding contralateral locations, by flipping the sign of the x-coordinate. Participant level parameter estimates (contrast values) will be extracted using MarsBar [138]. This approach uses the methods presented in a recent analysis of the CRUNCH effect in a similar population [40]. Correction for multiple comparisons will use the false discovery rate across the forty ROIs [139]. Secondary, exploratory analyses of the more general semantic control network will use the maps of semantic control for domain general control as identified in the [24] metaanalysis.

## Supporting information

**S1 File.**
(DOC)

## Acknowledgments

We wish to thank Perrine Ferré for sharing statistical maps that were used for power analysis.

## Author Contributions

**Conceptualization:** Niobe Haitas, Maximiliano Wilson, Yves Joanette.

**Data curation:** Niobe Haitas, Jason Steffener.

**Formal analysis:** Niobe Haitas, Mahnoush Amiri.

**Funding acquisition:** Yves Joanette.

**Investigation:** Niobe Haitas.

**Methodology:** Niobe Haitas, Mahnoush Amiri, Maximiliano Wilson, Jason Steffener.

**Software:** Jason Steffener.

**Supervision:** Maximiliano Wilson, Yves Joanette, Jason Steffener.

**Writing – original draft:** Niobe Haitas.

**Writing – review & editing:** Yves Joanette, Jason Steffener.

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
