## [Decision Letter · Decision Letter 0]

22 Dec 2020

PONE-D-20-27105

Age-preserved semantic memory and the CRUNCH effect manifested as differential semantic control networks: an fMRI study

PLOS ONE

Dear Dr. Haitas,

Thank you for submitting your manuscript to PLOS ONE. After careful consideration, we feel that it has merit but does not fully meet PLOS ONE’s publication criteria as it currently stands. Therefore, we invite you to submit a revised version of the manuscript that addresses the points raised during the review process.

The reviewers think that the proposed study may contribute to better understanding of cognitive compensation. However, they expressed multiple concerns regarding the theoretical framework, study design, statistical analysis, power calculation, and inclusion criteria. These concerns are summarized below and should be addressed before the manuscript is considered for publication

Theoretical framework:

The reviewers found it confusing that the Compensation Related Utilization of Neural Circuits Hypothesis (CRUNCH) focuses on task demands rather than age, but all proposed analyses propose to compare old and young adults for easy vs. difficult tasks that are not adjusted based on performance. From this perspective, it is unclear how ‘preservation’ of semantic memory is going to be tested in older adults because they are expected to show a decline in performance for a more difficult condition compared to young participants. Several reviewers pointed out that the differences between “perceived” and “actual” task difficulty should be examined and the implications for behavioral and neuroimaging results should be discussed in more detail. Given that the CRUNCH model explanation of the decreases in brain activation for more difficult tasks is only one possibility of several, the reviewers would appreciate you discuss alternative explanations for this phenomenon. In order to illustrate your hypotheses regarding RT, accuracy and brain activation change across levels of difficulty in younger and older adults, you might consider adding a figure that schematically illustrate these relationships.

Study design:

The ‘Stimuli description’ suggests that the task was piloted or validated already (p.14) to create the stimuli. It would help if you provided more detail on the age and other demographic factors of the sample that was tested to create the stimuli. How the semantic space associations were different for older vs. younger adults? Given that the proposed task was not validated against the CRUNCH model (at least no related information was reported in the manuscript), the reviewers were unclear about the correspondence between the U-shaped curve and task difficulty for older participants. Please provide basis for your hypothesis that older participants will be at the descending part of the U-shaped curve for difficult condition. Also please clarify how you are going to examine the U-shaped curve with only 2 levels of difficulty. Ideally, one would want at least 4 (better more) levels of difficulty to examine this kind of relationship.

Statistical Analyses and power calculation:

The reviewers pointed out that older and younger participants may have different HDR. Please indicate how you are going to account for these differences. The reviewers were also confused about whether you are going to conduct the analyses in the whole brain or in the ROIs. If you propose the ROI analysis, please give a more detailed description of how ROIs are going to be defined and how you are going to account for multiple comparisons across ROIs as well as across multiple task conditions and measures. Please describe specific statistical analyses that will be conducted for behavioral data and the fMRI-behavior relationships. It would be very helpful if you could give a definition of optimal and sub-optimal performance in the context of your specific study. For example, you propose that participants with a high error rate (outliers) will be excluded from further analysis but expect that old participants will have sub-optimal performance. How does the error rate correspond to ‘sub-optimal’ performance in the proposed study? The reviewers would like to see more detailed description of power calculation using *fmripower*. What pilot fMRI data were used as inputs and what were the expected effect size and alpha level?

Inclusion criteria:

Given that older participants may have a range of systemic illnesses due to age, please clarify how you are going to control for endocrine disorders, high blood pressure, medications, and other health issues in these individuals.

We look forward to receiving your revised manuscript.

Kind regards,

Anna Manelis, Ph.D.

Academic Editor

PLOS ONE

Reviewers' comments:

Reviewer's Responses to Questions

**Comments to the Author**

1. Does the manuscript provide a valid rationale for the proposed study, with clearly identified and justified research questions?

Reviewer #1: Partly

Reviewer #2: Partly

Reviewer #3: Partly

Reviewer #4: Yes

2. Is the protocol technically sound and planned in a manner that will lead to a meaningful outcome and allow testing the stated hypotheses?

Reviewer #1: Yes

Reviewer #2: Yes

Reviewer #3: No

Reviewer #4: Partly

3. Is the methodology feasible and described in sufficient detail to allow the work to be replicable?

Reviewer #1: Yes

Reviewer #2: Yes

Reviewer #3: No

Reviewer #4: Yes

4. Have the authors described where all data underlying the findings will be made available when the study is complete?

Reviewer #1: Yes

Reviewer #2: Yes

Reviewer #3: Yes

Reviewer #4: Yes

5. Is the manuscript presented in an intelligible fashion and written in standard English?

Reviewer #1: Yes

Reviewer #2: Yes

Reviewer #3: Yes

Reviewer #4: Yes

6. Review Comments to the Author

You may also provide optional suggestions and comments to authors that they might find helpful in planning their study.

Reviewer #1: Haitas et al. proposed to investigate the preservation of semantic memory in old age using a semantic judgment paradigm during fMRI. The primary objective is to test the compensation-related Utilization of Neural Circuits Hypothesis (CRUNCH) by comparing the differential effects of changing task demands on neural activation in young and older participants. Though testing of the popular theory in regard to semantic memory could be a valuable addition to the existing literature, I have two major and few minor concerns, including some clarification questions.

Major concerns:

1). Though the topic and aim of the study claim to explain the ‘preservation’ of semantic memory in old age but at the same time, the authors propose to do that according to CRUNCH. This is contradictory, as CRUNCH indicates a decline in performance in the face of a difficult task, in older adults, secondary to the exhaustion of compensatory neural activity. More specifically, the authors hypothesize that younger adults will surpass the older ones in performance when presented with a task of high difficulty (page 9, line 196). If the decrease in performance is expected, how could you justify the idea of ‘preservation’ in older folks?

2). In the abstract (page 3, lines 22-24), it says, “this study aims to test the Compensation Related Utilization of Neural Circuits Hypothesis (CRUNCH) focusing on task demands rather than age as a possible framework” but all the hypotheses of the study are based on condition-wise comparisons between the two age groups (page 9). It is suggested to make the objective of the study congruous with its research plan.

Minor concerns:

1). Page 3, line 64; The statement is inconsistent with the cited literature. The concept of CRUNCH proposed by Reuter-Lorenz et al. was not irrelevant to aging. The proper reference supporting the statement could be a seminal study by Schneider-Garces et al., 2010

Schneider-Garces, N.J., Gordon, B.A., Brumback-Peltz, C.R., Shin, E., Lee, Y., Sutton, B.P., Maclin, E.L., Gratton, G., Fabiani, M., 2010. Span, CRUNCH, and beyond: working memory capacity and the aging brain. J Cogn Neurosci 22, 655-669.

2). Throughout the paper, the authors use the term “CRUNCH-like”. CRUNCH is a hypothesis, and such wording is not accurate. Use of a better word (e.g., compensatory) is suggested.

3). Page 4, lines 76-80; The mentioned figure in the cited paper does not show a relationship between task demands and compensation but between task demand and fMRI activity to illustrate the concept of compensation. Please correct the wording throughout the paper.

4). Page 4, line 79, did the authors mean “a decrease in activity at higher task demands level”? a typo, maybe?

5). Pages 5&6, lines 107-109; Please clarify what age-range participants are being referred to.

6). Page 6, lines 116-119; This is not correct. The cited meta-analysis did not analyze the accuracy, they did measure performance in terms of reaction time, and the overall findings were more inclined to a decrease in performance in older adults as compared to the younger ones.

7). Page 7, lines 134-138; Please mention the age-range of the population of interest in the cited paper, as that information is highly relevant to the current proposal.

8). Page 8, lines 157-160; Here again, the age-range of the population studied in the cited literature needs to be mentioned.

9). Page 8, line 165; It's not clear what the authors mean by ‘stable’. Previously the authors supported the idea of compensatory neural activity in old age when performing simple tasks, but now here they are speculating that neural correlate of semantic memory may remain ‘stable’ with aging when performing simple tasks and only change when faced with a difficult one? This discrepancy in statements needs to be addressed.

10). Page 9, lines 187-188; Consider mentioning if the left/right lateralized or bilateral effects are expected in the regions of interest.

11) Page 10, lines 204-208; The hypothesis stated is not in line with CRUNCH. According to CRUNCH, in high demand condition, the compensatory activity exhausts in older adults. Secondly, please clarify what is meant by ‘higher level of task performance? Increased accuracy? decreased reaction time? or both?

12). Page 12; As the proposal suggests that the cohort of older people would have an age-range of 60-75 years. How would the common comorbidities like diabetes and hypertension be controlled? As both of these conditions are known to cause cognitive deficits. Understandably, the exclusion of people with such common comorbidities may make a goal of 40 participants challenging to reach. In such a case, is there any consideration for collecting the relevant information and controlling the effect of such confounders at the back end?

13). Page 12; Would any other relevant neuropsychological tests (e.g., semantic and episodic memory test, verbal fluency, etc.) be administered besides MoCA?

14). What is the statistical plan for behavioral effects? Also, need to mention statistical tests that would be used to probe the relation between behavioral and neural findings. Though the authors say that an exploratory analysis would be performed to investigate that, a little statistical detail would be helpful for the clarity of the planned analyses.

Reviewer #2: This proposal aims at investigating whether semantic control in younger and older adults follows the pattern predicted by the CRUNCH model. Participants will be performing a semantic decision task whereby they should pick one of two words that they think is more closely related to a target word. The semantic distance between the two choice words and the target word will either be great or small, producing “easy” and “difficult” conditions, respectively. The prediction is that in the easy condition, older adults will exhibit compensatory brain activity with maintained behavioral performance. However, in the “difficult” condition, brain activity will decline and behavioral performance will suffer.

I think this research has good potential, and I’m generally in favor of supporting this project. However, I also do have some concerns and recommendations that should be taken care of before the main project is initiated. I outline these below:

1. Theoretical rigor: Although the CRUNCH models provides a reasonable explanation for compensatory brain activity at low levels of task difficulty, its explanation for high levels of task difficulty is not very convincing (at least to me). Why should brain activity decline for difficult tasks? Why not just plateau? If an increase in brain activity is supposed to compensate for weakened cognitive ability, one would expect brain activity to increase up to a ceiling level and stay at that level. Behavioral performance cannot further improve when the ceiling brain activity is reached. One alternative possibility for why brain activity (and behavior) declines for difficult tasks is that processing becomes “good-enough” or “shallow” to save processing resources (e.g., Ferreira et al., 2002; Ferreira & Patson, 2007; Karimi & Ferreira, 2016). In other words, at high levels of task difficulty, older adults may resort to strategic performance. Such strategic processing may lead to a decrease in brain activity because participants may essentially bypass some aspects of the task. This is more likely for older adults because they don’t have the neural reserve to attend to all aspects of the task. In the current proposal, this may translate into making a quick decision based on whatever information becomes available first, rather than evaluating all the semantic features of the word before making a decision. I recommend that the authors go beyond CRUNCH to explain the expected pattern of results, especially for the “difficult” condition.

2. Confirmatory data via eye-tracking: This is just a suggestion and I don’t know if the authors have an MRI-compatible eye-tracker. But just in case they do, it would be nice to show that participants fixate on the two choice words more often (by going back and forth between them more often) in the difficult relative to the easy condition. Then, if older adults do any good-enough/shallow processing, their eye-movements should reveal this; they should look at the choice words less often in the difficult relative to the easy condition, but younger adults may show the opposite pattern.

3. Apparent vs. actual difficulty: It is not currently clear how “apparent” and “actual” difficulty may affect the results. If eye-tracking is not possible, the authors can explicitly ask the participants how difficult the current item was after each trial. This way, they can run another analysis on the “participant-rated” easy and difficult conditions and see if the results differ.

4. Neuropsychological tests: Currently, the neuropsychological tests are very limited. How can we ensure that the results are not caused by cognitive abilities other than semantic control, including inhibition, attention, speed of processing, working memory capacity, language knowledge etc.? I recommend that that author collect data on these individual differences and analyze the data accordingly.

5. Feature vs. association: It is currently unclear if the semantic distance between words is affected by semantic features or semantic associations. Because associations may give rise to stronger automatic cognitive processes such as semantic priming, and because such automatic processing are largely preserved in older adults, I recommend the author try to minimize strongly-associated words and focus more on shared semantic features to increase power.

Reviewer #3: This study investigates the influences of task demands on the brain activity serving semantic memory, based on the CRUNCH model. To conduct the study, the authors plan to manipulate the levels of task demands and compare the corresponding differences in neurofunctional activation in young and old age groups. Several expectations are made by the authors: 1) the activation in semantic control regions will be significantly different between young and old participant groups. Specifically, in the low-demand condition, older adults will present significant increases in activation in left-hemisphere control regions compared to the younger participants, reflecting the compensation for higher perceived task difficulty. 2) In the high-demand condition, younger adults will present higher activation in the semantic control regions than older adults, reflecting the compensational overactivation in the older group has reached a threshold beyond which additional neural resources do not suffice. 3) the behavioral performance will also follow the inverted U-shaped relation between compensation and task demands.

Comments

The motivation of this study is clear and the relevant background literature is covered at an appropriate level of detail. It has potential to make some contribution to the understanding of the neural basis of cognitive aging in semantics. However, this paper is difficult to follow in places and the logic and rationale for the authors’ hypotheses are unclear, so I feel some clarification should be made. Additionally, I have a number of concerns about the details of the experimental design and data analysis which need to be better articulated to understand what the authors are proposing.

First of all, the logic and rationale for the authors’ three hypotheses are unclear (Page 9). Based on the CRUNCH model, semantic control regions increase their activations to compensate for increased task demands, whereby after a certain difficulty threshold, available neural resources have reached their maximum capacity and further demand increases lead to reduced activations. However, it’s not clear how they’ve translated this verbally described theory into experimental predictions in hypotheses 1 to 3. Why do they believe, for example, that their high-demand condition will catch older participants at the descending part of the CRUNCH curve and not, for example, on the ascent or at the plateau? It’s not clear to me how they can be confident about this. Do they, for example, have some behavioral data indicating that older people are at ceiling in the low-demand condition but are more error-prone in the high-demand condition? This would help to reassure me that older people will have exceeded their processing limits in the high-demand condition, particularly if young people showed a different pattern.

The concept of “perceived task difficulty” in different age groups is introduced to help establish the authors’ hypotheses, but the relation between “perceived difficulty” and actual task demand/difficulty and “perceived difficulty” fits in the CRUNCH theory are not well discussed. Additionally, the authors assume that by manipulating the two levels of task demands (low and high) they can precisely locate the “perceived difficulties” in the increasing and descending part of the inverted U-shaped model, I feel this is not that easy to achieve.

I also found the methods of data analysis unsatisfactory and lacking in detail.

First, hypotheses seem under-specified:

Hypothesis 1 predicts a demand x age interaction in accuracy and RT and in BOLD activation in 6 separate ROIs (pg. 9). These requires 8 statistical tests. To conclude that the hypothesis is upheld, do all of these effects need to be significant? Or only some of them? If so, which ones? Confusingly, in the data analysis section it’s stated that this hypothesis will be tested at the whole brain level, contradicting the description provided earlier in the paper.

Hypothesis 2 predicts that in the low-demand condition, participants will make minimal errors. How will this be tested? What is the definition of minimal errors? It is also hypothesized the older people will present longer RTs and more activation in control regions. Again, how many tests have to be significant to accept this hypothesis?

Hypothesis 3 predicts that in the high-demand condition, only young adults will perform optimally (again, defined how?), that they will perform better than older people in accuracy and RT, that older people will show less activation in left-hemisphere semantic control regions (all 6? Or is just 1 enough?) and more activation in right-hemisphere homologues (again, 6 tests here?).

Second, critical information is lacking about the ROIs. There are 12 ROIs in total in this study (Page 22, 23), but very little information is provided about how they are defined and how they will treated statistically. The Neuromorphometrics atlas that the authors want to use to define the ROIs does not include some of the specific regions they discuss. For example, this atlas does not have a single IFG area. How will the various IFG subregions in this atlas be combined to make a single ROI? It does not have an area labelled dorsal inferior parietal cortex. How will this be defined? It does not divide the MTG into anterior and posterior parts, so how will pMTG be defined? And so on.

In addition, as stated earlier, the same hypotheses are being tested in multiple ROIs. The authors should state how they are handling issues of multiple comparisons as some sort of correction seems necessary here.

Finally, more information should be given on the discussion of power calculations (Page 11, 12). The authors mention that “power estimates calculated that a sample of 38 participants per age group would be sufficient to detect age group differences bilaterally in the IFG, AG, dACC, dIPC, and large areas of the PFC”. What are the expected effect sizes in each of these regions, what alpha level has been used to estimate power (and does it account for multiple comparisons?) and what power is achieved with this sample size?

Minor comments

Page 9. The sentences of the three hypotheses are long and difficult to understand. For example, the paper says: “If this interaction is not found between task demands and age, it is expected that … (Line 190)”. The causal relationship between hypothesis 1 and 2/3 that the authors are claiming here is not clear.

Page 16. The estimation of the duration of the fMRI experiment does not add up. There are two runs in total in this experiment, and the duration for each run is 9:45 minutes. The authors suggest that the fMRI session will take 90 minutes, which is much longer than I expected even considering all the related preparation procedures.

Reviewer #4: Haitas et al. present a protocol for a registered report for an fMRI experiment testing the CRUNCH model of cognitive compensation. They propose to compare 40 young and 40 older adults on a semantic judgement task. In general, the experiment is well described, and the manuscript is clearly written. I am a little concerned that perhaps the design requires some additional consideration, and I outline this below. I welcome this registered report examining the CRUNCH model as numerous studies in this area are under-powered and poorly designed; as such I believe this work may have a substantial impact in this area.

MAJOR COMMENTS

1. I am surprised that the experimental design only seems to include 2 levels of task difficulty, when the literature states that 3-4 levels of difficulty (Cappell et al.; Reuter-Lorenz & Cappell; Fabiani et al.) is required to determine the hypothesised non-linear relationship between task difficulty and brain activity. The authors refer to this non-linear relationship in the introduction (‘inverted U-shaped one’, page 4). How can this experimental design adequately test the CRUNCH model with only 2 levels of difficulty? I would recommend adding at least a third level of difficulty to the experimental design.

2. A few points about the fMRI data analysis:

a. it is becoming much more common to include derivatives of the motion parameters in the GLM as well as the motion parameters themselves.

b. Also, is there a reason why HDR derivatives are not included in the model? Older participants may have different HDR shape/timing compared to younger people.

c. I also strongly suggest accounting for confounds in the model – at the very least, accounting for grey matter/cortical thickness differences between the groups, if not other physiological parameters like cardiovascular parameters, sex, haemoglobin, etc. Differences in grey matter volume between the groups should be explicitly tested for and controlled in the analysis.

d. Also, is the ‘baseline’ implicit baseline or the control task? Is the post-hoc tests of age group (pg 22 ln 469) a t-test? How does this control for potential cardiovascular confounds between the age-groups?

e. I am a little confused about whole brain vs. ROI analysis. Most of the analysis section seems to be referring to a whole brain approach but then the hypotheses are focused on the ROIs. Which parameters will be calculated from the ROIs? How will they be analysed?

f. I think that if the analysis approach is modified (e.g., including derivatives of HDR & motion parameters, grey matter, etc.) then the power calculation should be revisited.

3. One of the reasons why I welcome this registered report is the recent study by Jamadar et al. in Neuropsychologica. These authors suggest that there may be publication bias in the reporting of the CRUNCH effect, and perhaps the effect is not as robust as believed. It would be worthwhile mentioning this criticism in the manuscript, maybe in the introduction.

4. “Power estimates calculated that a sample of 38 participants per group would be sufficient to detect age group differences bilaterally …” does this account for multiple comparisons correction? Could the authors provide an alpha level here, and note how they will correct for multiple comparisons across the 10 (?) ROIs. Also, what does ‘large areas of the PFC’ mean? Is this a large ROI across subsections of the PFC?

5. Stimuli description – I didn’t quite follow this section. Have the authors already run a pilot in 100 people to collect the semantic relationship data? Are these people similarly aged as the experimental groups? I could imagine that a semantic relationship might exists for one age group and not another (e.g., stream – music), have the authors assessed if the semantic relationships are maintained across age groups? How many participants were excluded on the basis of the correlation < 0.6 threshold? Is there behavioural data to confirm that ‘hard’ trials are actually difficult, and ‘easy’ trials are actually easy – and comparable between the age groups? One of the challenges of the CRUNCH model is the definition of ‘difficulty’ is not as clear cut in all cognitive domains as it is in the memory domain, and I think that it is worthwhile confirming that task difficulty is working as expected, and similarly across age groups.

MINOR COMMENTS

1. A figure would be useful to illustrate the hypotheses.

2. The motion exclusion strategy is fairly conservative. Are the parameters given (2mm/1deg) acute motion or drift across time? Perhaps there may be more data loss for the older than younger participants with this criterion.

3. The TR is quite long – is there a reason for this? And is there a reason why the EPI resolution is not isotropic?

4. Participants respond with ‘any of their fingers’ – it is typical to standardise the response keys across subjects.

5. I think that the activation height threshold noted on pg 22 ln 478 needs ‘uncorrected’ specified.

6. “Participants who qualify for the fMRI scanning session…” do participants need to meet a certain threshold of performance in the 15 practice triads to qualify? Does ‘qualify’ here mainly mean ‘meet inclusion criteria’? This is unclear. Also will session 2 always be one week later, or will there be a window of opportunity.

7. PLOS authors have the option to publish the peer review history of their article (what does this mean?). If published, this will include your full peer review and any attached files.

Reviewer #1: No

Reviewer #2: **Yes: **Hossein Karimi

Reviewer #3: No

Reviewer #4: No

---

## [Author Response · Author response to Decision Letter 0]

5 Mar 2021

PONE-D-20-27105

Age-preserved semantic memory and the CRUNCH effect manifested as differential semantic control networks: an fMRI study

PLOS ONE

Dear Dr. Haitas,

Thank you for submitting your manuscript to PLOS ONE. After careful consideration, we feel that it has merit but does not fully meet PLOS ONE’s publication criteria as it currently stands. Therefore, we invite you to submit a revised version of the manuscript that addresses the points raised during the review process.

The reviewers think that the proposed study may contribute to better understanding of cognitive compensation. However, they expressed multiple concerns regarding the theoretical framework, study design, statistical analysis, power calculation, and inclusion criteria. These concerns are summarized below and should be addressed before the manuscript is considered for publication

Theoretical framework:

The reviewers found it confusing that the Compensation Related Utilization of Neural Circuits Hypothesis (CRUNCH) focuses on task demands rather than age, but all proposed analyses propose to compare old and young adults for easy vs. difficult tasks that are not adjusted based on performance. 

Thank you for your critical and constructive comments.

To account for performance, brain imaging analysis will focus on correct trials only ensuring that we are looking at brain activation related to accurate performance. Analyses will explicitly focus on the relationships between brain activation and task performance. These analyses will identify brain regions where age group differences in activation are dependent or independent of task performance. 

From this perspective, it is unclear how ‘preservation’ of semantic memory is going to be tested in older adults because they are expected to show a decline in performance for a more difficult condition compared to young participants. 

Indeed, there seems to be a contradiction when we refer to well-maintained semantic memory and then aim to test a hypothesis that suggests that older adults will perform worse than younger ones in a semantic judgment task. It is generally thought that in healthy aging, performance in tasks that require attention and control decrease, whereas tasks that depend on lifelong learning (such as semantic memory) are typically well maintained (Fabiani, 2012). Thus, when compared with other cognitive functions in aging (e.g. attention, memory), semantic memory is well preserved (Hoffman & Morcom, 2018; Salthouse, 2004) (see e.g. (St-Laurent, Abdi, Burianova, & Grady, 2011) comparing autobiographical, episodic and semantic memory in young vs. older adults). When comparing semantic memory of older with younger adults, the literature has yielded numerous results in regards to neural activation increases or decreases, and performance (accuracy and response times), depending on the task utilized, inter-individual variability and the specific age group. For example, older adults are found to be performing equally to younger adults in semantic priming tasks (Allen, Madden, Weber, & Groth, 1993; Laver, 2009; Lustig & Buckner, 2004) or tasks assessing vocabulary size (Krieger-Redwood et al., 2019). However, more ‘tip of the tongue’ phenomena are reported for older adults (Diaz, Rizio, & Zhuang, 2016), reduced performance in older adults in a naming task (Verhaegen & Poncelet, 2013) and increased response times when tasks involve semantic selection (Hoffman & Morcom, 2018). The answer may lie within the system of semantic memory which is thought to comprise of 2 or 3 sub-systems, and with each one of them differentially affected by aging. More specifically, a) semantic representations are thought to be well-maintained in aging, b) retrieval processes are thought to be age-invariant whereas c) semantic control processes are thought to be negatively impacted by aging (Hoffman 2018). 

In the manuscript, it is reported: Performance in terms of accuracy in semantic tasks is generally well maintained in older adults considering their more extensive experience with word use and a larger vocabulary than younger adults (Balota, Cortese, Sergent-Marshall, Spieler, & Yap, 2004; Kahlaoui et al., 2012; Kavé, Samuel-Enoch, & Adiv, 2009; Laver, 2009; Methqal, Marsolais, Wilson, Monchi, & Joanette, 2018; Verhaegen & Poncelet, 2013; Wingfield & Grossman, 2006). Response times (RTs) however are often longer compared to younger adults (Balota et al., 2004), possibly because older adults are slower in accessing and retrieving conceptual representations from their semantic store (Bonner, Peelle, Cook, & Grossman, 2013; Huang, Meyer, & Federmeier, 2012; Wierenga et al., 2008), engaging the required executive function resources (Diaz, Johnson, Burke, & Madden, 2018), and necessary motor responses (Falkenstein, Yordanova, & Kolev, 2006). 

In terms of brain activation, semantic memory is overall well-maintained throughout aging, as the neural correlates sustaining it are reported to be largely age-invariant, with only small differences existing in neural recruitment as a function of age (Hoffman & Morcom, 2018; Kennedy114 et al., 2015; Lacombe, Jolicoeur, Grimault, Pineault, & Joubert, 2015). Other studies report that even if semantic memory performance is equivalent between younger and older, the neural circuits that support it are different between the two groups (Wierenga 2008, Federmeier 2007). Similar findings of different neural circuits despite age-matched performance were found on working memory tasks, indicating that older adults may achieve the same outcomes using different neural circuits or strategies for example (Cappell, Gmeindl, & Reuter-Lorenz, 2010). However, despite semantic memory performance (e.g. accuracy) being well preserved in aging when compared to other cognitive functions, this preservation can be potentially affected by increased task demands, as demonstrated for example in studies that have manipulated levels of demand in semantic memory tasks (Chiou, Humphreys, Jung, & Lambon Ralph, 2018; Davey et al., 2015; Kennedy et al., 2015; Noppeney & Price, 2004; Sabsevitz, Medler, Seidenberg, & Binder, 2005; Zhuang, Johnson, Madden, Burke, & Diaz, 2016). Semantic memory of younger adults as well is affected by increasing task demands (Badre, Poldrack, Paré-Blagoev, Insler, & Wagner, 2005; Krieger-Redwood, Teige, Davey, Hymers, & Jefferies, 2015; Noonan, Jefferies, Visser, & Lambon Ralph, 2013). To reflect the above, we have edited some parts of the manuscript to emphasize the contradiction of well-maintained semantic memory e.g. maintained representations but affected semantic control in aging, as well as performance being affected by increasing task demands. 

Several reviewers pointed out that the differences between “perceived” and “actual” task difficulty should be examined and the implications for behavioral and neuroimaging results should be discussed in more detail. 

In regards to perceived task difficulty, an additional session with participants one week following the fMRI acquisitions will take place, whereby they will rate each triad on a difficulty 1-7 likert scale (eg. 1: very easy, 7: very difficult). We will further assess whether perceived difficulty correlates with actual performance scores (accuracy rates and RTs) and whether perceived difficulty correlates with levels of activation in the young and older adults e.g. whether increased levels of perceived difficulty correlate with increased RTs and reduced accuracy, as well as levels of activation in semantic control regions. 

Given that the CRUNCH model explanation of the decreases in brain activation for more difficult tasks is only one possibility of several, the reviewers would appreciate you discuss alternative explanations for this phenomenon.

Additional theoretical frameworks were introduced as an alternative to the CRUNCH framework to justify neurofunctional differences between older and younger participants (page 4). These include the good-enough model suggested by reviewer #2 (Karimi & Ferreira, 2015), the incapacity to deactivate the default mode network (Humphreys 2015, Vatansever 2017, Persson 2007), the HAROLD (Cabeza, 2002), PASA (Davis, Dennis, Daselaar, Fleck, & Cabeza, 2008) and cognitive reserve (Stern, 2009) hypotheses. We included also critiques that have been voiced for the above models, (Berlingeri, Danelli, Bottini, Sberna, & Paulesu, 2013, Li et al. 2015, Festini 2018) and also for CRUNCH (Jamadar, 2020). As the neuroimaging and aging literature is quite vast, we chose to focus on the CRUNCH compensatory framework to test it on semantic memory since it is more general, it accounts for individual differences and makes predictions within the individual, when compared with HAROLD or PASA for example (Festini 2018). 

In order to illustrate your hypotheses regarding RT, accuracy and brain activation change across levels of difficulty in younger and older adults, you might consider adding a figure that schematically illustrate these relationships.

Three figures were added to illustrate our hypotheses on brain activation, accuracy and RTs differences between young and older across different levels of task difficulty.

Study design:

The ‘Stimuli description’ suggests that the task was piloted or validated already (p.14) to create the stimuli. It would help if you provided more detail on the age and other demographic factors of the sample that was tested to create the stimuli. How the semantic space associations were different for older vs. younger adults?

Information from the stimuli pilots was added in the section ‘Stimuli description’ page 36 of the manuscript, showing that there was a significant main effect of task for accuracy and RTs and a significant effect of age group for RTs. 

Given that the proposed task was not validated against the CRUNCH model (at least no related information was reported in the manuscript), the reviewers were unclear about the correspondence between the U-shaped curve and task difficulty for older participants. Please provide basis for your hypothesis that older participants will be at the descending part of the U-shaped curve for difficult condition. Also please clarify how you are going to examine the U-shaped curve with only 2 levels of difficulty. Ideally, one would want at least 4 (better more) levels of difficulty to examine this kind of relationship.

Several papers in the literature have referred to the inverted U-shape relation between task demands and fMRI activation (Rypma 2007, Cabeza et al., 2018; Reuter-Lorenz & Cappell, 2008), and this has inspired our research aims. However, our aim is not to test the validity of the CRUNCH model or the validity of the inverted U-shape itself. This would indeed require more task demand level points. Semantic memory on the other hand is context-dependent and creating task demand levels for semantic memory would be complex. Instead, we aim to test the predictions of the CRUNCH model regarding differences between age groups. These are expressed as: 1) age related over-activation at low loads 2) age-related under-activation at high loads (Cappell et al., 2010). These predictions can be evaluated with 2 levels of task demands only. Our second focus is to test these predictions as manifested specifically in the semantic control network (ROIs).

More specifically, the CRUNCH model was conceived on a working memory task of a quantifiable nature (number of items to retain) (Cappell et al., 2010). Studies that have tested CRUNCH with more than 2 difficulty levels, have done so on working memory ((Fabiani, 2012; Rypma, Eldreth, & Rebbechi, 2007; Schneider-Garces et al., 2010), including recently an evaluation of the model with 4 demand levels on working memory (Jamadar, 2020). Our focus is on semantic memory which is of a different nature, more context-dependent and more difficult to quantify or manipulate for task demands (see for example Patterson et al. 2007 on the salience of semantic features and Badre et al. 2002 on the variability of strength between semantic representations). There is no previous study in our knowledge which has evaluated the CRUNCH model in the context of semantic memory with more than two levels of task demands (see however the behavioral study of Fu et al. 2017). Studies on semantic memory that have examined the impact of differing task demands, have done so with 2 levels (Chiou et al., 2018; Davey et al., 2015; Kennedy et al., 2015; Noppeney & Price, 2004; Sabsevitz et al., 2005; Zhuang et al., 2016). 

Testing the validity of the CRUNCH inverted-U shape on semantic memory with 4 levels of task demands would require a very complex methodology. For example, Jamadar (2020) states: ‘the CRUNCH model is challenging to test because it must be possible to manipulate task difficulty, parametrically, across 3–4 levels. Not all cognitive constructs/tasks are amenable to these requirements’. The stimuli that was developed for the current study (360 words overall) was developed following numerous iterations, evaluations from pilots and peer-reviewed by a team of linguists. Words have numerous connotations and can elicit varying semantic representations. Factors such as frequency, imageability, length, age of acquisition, familiarity, emotional valence, concreteness (among many other factors) interplay with task demands and can thus influence processing of these representations (see for example, Sabsevitz, Binder 2011, Meteyard 2012, Barsalou 2008).

In regards to our stimuli and the required difficulty level to provoke a differential activation between the 2 task demand levels, 1) by design, there is an added level of complexity, provided within the definition of conditions (low-high) and 2) our behavioral data from pilots with 28 participants provide evidence of a significant main effect of task for both accuracy and RTs. Additionally, to account for individual differences of difficulty between participants, we have included in the manuscript: ‘Subsequent analyses will explore this question with heterogeneous slopes models using individualized rescaled levels of task difficulty and will compare brain activation with performance, brain activation with perceived difficulty and performance with perceived difficulty. using a throughput measure (Schneider-Garces et al., 2010). This approach will determine how the relationship between individual task difficulty and brain activity is affected by age group’. 

As such, our goal is to test the main predictions of the CRUNCH framework as first evidence of its applicability to semantic memory, manifested with differential activation in the semantic control network. We have rephrased these sentences in our hypotheses to better explain that we focus on the predictions rather than the validity of the whole CRUNCH model. 

Statistical Analyses and power calculation:

The reviewers pointed out that older and younger participants may have different HDR. Please indicate how you are going to account for these differences. 

We will account for differences in HDR between younger and older adults as such (p. 45): The event-related first-level statistical model of the fMRI data will include the event-chain convolved with the double-Gamma hemodynamic response function and its first derivative. The inclusion of this extra regressor will capture variance in the data due to any inter-participant or inter-group variations in the shape of the hemodynamic responses. 

The reviewers were also confused about whether you are going to conduct the analyses in the whole brain or in the ROIs. If you propose the ROI analysis, please give a more detailed description of how ROIs are going to be defined and how you are going to account for multiple comparisons across ROIs as well as across multiple task conditions and measures. 

The analyses to address the primary hypotheses of this study will be done within ROIs. The description of the ROI definition and analyses, as well as correction for multiple comparisons are now revisited in the ‘Defining the anatomical/functional ROIs’ section p.46. 

Please describe specific statistical analyses that will be conducted for behavioral data and the fMRI-behavior relationships. 

Behavioral data (RT and accuracy) will be analyzed using mixed- design ANOVA with age as a between-subjects factor and condition (high vs. low demands) as within-subject factor. Accuracy rates will be transformed using Fisher logit approximation to avoid ceiling effects.

Group analyses of the imaging data will be performed including behavioral covariates to investigate age group differences in the relationships between brain activity and task performance. Multiple comparisons across the 40 ROIs will be made using false discovery rate adjustments.

It would be very helpful if you could give a definition of optimal and sub-optimal performance in the context of your specific study. For example, you propose that participants with a high error rate (outliers) will be excluded from further analysis but expect that old participants will have sub-optimal performance. How does the error rate correspond to ‘sub-optimal’ performance in the proposed study? 

The terms ‘optimal’ and ‘sub-optimal’ are confusing. We replaced them with ‘better’ and ‘worse’ respectively, as performance is planned to be defined from the comparison between the young and the older adults. In regards to outliers, these are defined as two standard deviations higher than group average, and will be excluded from the analysis.

The reviewers would like to see more detailed description of power calculation using fmripower. What pilot fMRI data were used as inputs and what were the expected effect size and alpha level?

The power analysis section was completely redone. The expected effect size is 90% and alpha level is 0.05. We uploaded the table of effect sizes of the Ferre et al. (2020) used as input for our power analysis at osf.io. 

Inclusion criteria:

Given that older participants may have a range of systemic illnesses due to age, please clarify how you are going to control for endocrine disorders, high blood pressure, medications, and other health issues in these individuals.

In the participants’ section (page 15), information had been shared on a health questionnaire /pre-screening to take place on the phone, with the aim to exclude participants with a history of illness, drug use and other health issues (including endocrine disorders, high blood pressure and medication use), as well as multilingualism. We uploaded this questionnaire at the osf.io platform (currently in French language).

● A rebuttal letter that responds to each point raised by the academic editor and reviewer(s). You should upload this letter as a separate file labeled 'Response to Reviewers'.

● A marked-up copy of your manuscript that highlights changes made to the original version. You should upload this as a separate file labeled 'Revised Manuscript with Track Changes'.

● An unmarked version of your revised paper without tracked changes. You should upload this as a separate file labeled 'Manuscript'.

We look forward to receiving your revised manuscript.

Kind regards,

Anna Manelis, Ph.D.

Academic Editor

PLOS ONE

Reviewers' comments:

Reviewer's Responses to Questions

Comments to the Author

1. Does the manuscript provide a valid rationale for the proposed study, with clearly identified and justified research questions?

Reviewer #1: Partly

Reviewer #2: Partly

Reviewer #3: Partly

Reviewer #4: Yes

2. Is the protocol technically sound and planned in a manner that will lead to a meaningful outcome and allow testing the stated hypotheses?

Reviewer #1: Yes

Reviewer #2: Yes

Reviewer #3: No

Reviewer #4: Partly

3. Is the methodology feasible and described in sufficient detail to allow the work to be replicable?

Reviewer #1: Yes

Reviewer #2: Yes

Reviewer #3: No

Reviewer #4: Yes

4. Have the authors described where all data underlying the findings will be made available when the study is complete?

Reviewer #1: Yes

Reviewer #2: Yes

Reviewer #3: Yes

Reviewer #4: Yes

5. Is the manuscript presented in an intelligible fashion and written in standard English?

Reviewer #1: Yes

Reviewer #2: Yes

Reviewer #3: Yes

Reviewer #4: Yes

6. Review Comments to the Author

You may also provide optional suggestions and comments to authors that they might find helpful in planning their study.

Reviewer #1: Haitas et al. proposed to investigate the preservation of semantic memory in old age using a semantic judgment paradigm during fMRI. The primary objective is to test the compensation-related Utilization of Neural Circuits Hypothesis (CRUNCH) by comparing the differential effects of changing task demands on neural activation in young and older participants. Though testing of the popular theory in regard to semantic memory could be a valuable addition to the existing literature, I have two major and few minor concerns, including some clarification questions.

Major concerns:

1). Though the topic and aim of the study claim to explain the ‘preservation’ of semantic memory in old age but at the same time, the authors propose to do that according to CRUNCH. This is contradictory, as CRUNCH indicates a decline in performance in the face of a difficult task, in older adults, secondary to the exhaustion of compensatory neural activity. More specifically, the authors hypothesize that younger adults will surpass the older ones in performance when presented with a task of high difficulty (page 9, line 196). If the decrease in performance is expected, how could you justify the idea of ‘preservation’ in older folks?

(see paragraph above -response to the editor):

2). In the abstract (page 3, lines 22-24), it says, “this study aims to test the Compensation Related Utilization of Neural Circuits Hypothesis (CRUNCH) focusing on task demands rather than age as a possible framework” but all the hypotheses of the study are based on condition-wise comparisons between the two age groups (page 9). It is suggested to make the objective of the study congruous with its research plan.

The incongruence was corrected in the abstract as such: ‘…focusing on task demands and aging as a possible framework…’ in order to more accurately depict the CRUNCH framework.

Minor concerns:

1). Page 3, line 64; The statement is inconsistent with the cited literature. The concept of CRUNCH proposed by Reuter-Lorenz et al. was not irrelevant to aging. The proper reference supporting the statement could be a seminal study by Schneider-Garces et al., 2010

Schneider-Garces, N.J., Gordon, B.A., Brumback-Peltz, C.R., Shin, E., Lee, Y., Sutton, B.P., Maclin, E.L., Gratton, G., Fabiani, M., 2010. Span, CRUNCH, and beyond: working memory capacity and the aging brain. J Cogn Neurosci 22, 655-669.

Thank you for your comment and recommending the reference. Indeed, CRUNCH is not irrelevant to aging, it emphasizes however that is the level of task demands that impacts the level of brain activation and this, at any age (Reuter-Lorenz & Cappell, 2008). The defining feature of the CRUNCH model is task demands whereas predictions are made within-person as a function of task demands and availability of neural resources (Festini 2018). The authors also suggest that ‘We also show that this age-related over- activation occurs in a region… that was also recruited by young adults at higher memory loads suggesting that increased recruitment of this regions is not an aberrant sign of aging, but may instead be a typical compensatory neural response to increased cognitive demand’ (Cappell et al., 2010). When we put the emphasis on task demands rather than aging we were aiming at ‘paralleling’ aging to increased task demands and emphasizing the resilience of the brain to face its own deficits, as expressed by CRUNCH, and later on elaborated by the same authors in the STAC model (Park & Reuter-Lorenz, 2009): ‘Scaffolding is a process that characterizes neural dynamics across the lifespan. It is not merely the brain’s response to normal aging; it is the brain’s normal response to challenge’, as well as in (Cabeza et al., 2018) where aging is parallelized to brain injuries, lesions or disorders: ‘That is, individuals with a neurological disease or disorder may compensate for their disorder-related deficits in ways similar to those described here for healthy older adults’. The CRUNCH framework is thought to apply to young adults as well, who would demonstrate increased compensatory activations once task demands exceed a certain level. To reflect the CRUNCH framework more accurately, the manuscript has been modified as per your recommendation and every part referring to ‘task-demands only’ was replaced by ‘task-demands and aging’.

2). Throughout the paper, the authors use the term “CRUNCH-like”. CRUNCH is a hypothesis, and such wording is not accurate. Use of a better word (e.g., compensatory) is suggested.

This was corrected accordingly throughout the manuscript such as ‘compensatory’ or ‘compatible with CRUNCH’.

3). Page 4, lines 76-80; The mentioned figure in the cited paper does not show a relationship between task demands and compensation but between task demand and fMRI activity to illustrate the concept of compensation. Please correct the wording throughout the paper.

This was corrected accordingly throughout the manuscript (compensation was replaced by fMRI activation in the context of the inverted-U shaped discussion).

4). Page 4, line 79, did the authors mean “a decrease in activity at higher task demands level”? a typo, maybe?

Indeed a typo, thank you for pointing out, this was corrected. 

5). Pages 5&6, lines 107-109; Please clarify what age-range participants are being referred to.

This was added in the revised manuscript. 

6). Page 6, lines 116-119; This is not correct. The cited meta-analysis did not analyze the accuracy, they did measure performance in terms of reaction time, and the overall findings were more inclined to a decrease in performance in older adults as compared to the younger ones.

Thank you for the diligent reading of the cited literature. However, we have a slightly different interpretation of the conclusions of the cited article, (Hoffman & Morcom, 2018) where it is stated: ‘Effect sizes were computed from number of correct responses/errors but not from reaction times, since older people exhibit general reductions in processing speed that may not reflect changes in semantic processing per se’.

7). Page 7, lines 134-138; Please mention the age-range of the population of interest in the cited paper, as that information is highly relevant to the current proposal.

The specific article mentioned (Reuter-Lorenz & Cappell, 2008) is a review article and does not specify age ranges of the younger-older populations. Previous work of the authors (Reuter-Lorenz, Stanczak, & Miller, 1999) mentions ‘Twenty-four healthy older adults (age 65-75 years) and 24 younger adults (age 18-25 years) participated…’. In the study (Cappell, Gmeindl, & Reuter-Lorenz, 2010) used for the 2008 paper mean age of young was 20.8 years old and of older 68.4 years old (Cappell, K., Gmeindl, L.,&Reuter-Lorenz, P.A. (2006, November) (Age differences in DLPFC recruitment during verbal working memory maintenance depend on memory load. Paper presented at the annual meeting of the Society for Neuroscience, Atlanta, GA)

8). Page 8, lines 157-160; Here again, the age-range of the population studied in the cited literature needs to be mentioned.

The (Cabeza et al., 2018) and (Cabeza & Dennis, 2009) are review rather than experimental articles and thus do not focus on specific populations but comment on previous research comparing younger with older adults. The (Hoffman & Morcom, 2018) metaanalysis states: The mean age of young participants was 26.0 years (SD=4.1) and the mean age of older participants was 69.1 (SD=4.7), which was added in the manuscript.

9). Page 8, line 165; It's not clear what the authors mean by ‘stable’. Previously the authors supported the idea of compensatory neural activity in old age when performing simple tasks, but now here they are speculating that neural correlate of semantic memory may remain ‘stable’ with aging when performing simple tasks and only change when faced with a difficult one? This discrepancy in statements needs to be addressed.

Thank you for your comment. The response is related to the one above about maintained semantic memory. Some studies report that the neural correlates sustaining semantic memory are largely age-invariant, with only small differences existing in neural recruitment as a function of age (Hoffman & Morcom, 2018; Kennedy114 et al., 2015; Lacombe, Jolicoeur, Grimault, Pineault, & Joubert, 2015). Other studies report that even if semantic memory performance is equivalent between younger and older, the neural circuits that support it are different between the two groups (Wierenga 2008). Similar findings of different neural circuits despite age-matched performance were found on working memory tasks, indicating that older adults may achieve the same outcomes using different neural circuits or strategies for example (Cappell, Gmeindl, & Reuter-Lorenz, 2010). However, despite semantic memory performance (e.g. accuracy) being well-preserved in aging when compared to other cognitive functions, this preservation can be potentially affected by increased task demands, as demonstrated in studies that have manipulated semantic memory task demands (Chiou, Humphreys, Jung, & Lambon Ralph, 2018; Davey et al., 2015; Kennedy et al., 2015; Noppeney & Price, 2004; Sabsevitz, Medler, Seidenberg, & Binder, 2005; Zhuang, Johnson, Madden, Burke, & Diaz, 2016). Semantic memory of younger adults as well is affected by increasing task demands (Badre, Poldrack, Paré-Blagoev, Insler, & Wagner, 2005; Krieger-Redwood, Teige, Davey, Hymers, & Jefferies, 2015; Noonan, Jefferies, Visser, & Lambon Ralph, 2013). To resolve the contradiction, the term ‘stable’ was replaced by ‘relatively age-invariant’, to reflect small only changes in activation.

10). Page 9, lines 187-188; Consider mentioning if the left/right lateralized or bilateral effects are expected in the regions of interest.

Thank you for your comment. There is a bias in semantic processing to recruit mainly left-lateralized regions. However, when task demands increase, activation tends to become more bilateral. Hence, the primary ROIs are bilateral and this was added in the manuscript.

11) Page 10, lines 204-208; The hypothesis stated is not in line with CRUNCH. According to CRUNCH, in high demand condition, the compensatory activity exhausts in older adults. Secondly, please clarify what is meant by ‘higher level of task performance? Increased accuracy? decreased reaction time? or both?

Thank you for your comment. Indeed, according to CRUNCH we would expect that compensatory activity exhausts in older adults in the high-demand condition. However, the CRUNCH model predicts variability in older adults’ performance and level at which they reach their personal ‘crunch’ point (peak level of the inverted-U curve), depending on cognitive reserve and perceived task difficulty (Lustig et al. 2009). We expect that if there are adults who perform better than their counterparts (reduced errors and RTs) they may not have exhausted their compensatory activity during the high-demand condition, in which case we could expect them to show activation similar to the young. The definition of higher level of task performance was clarified in the manuscript. 

12). Page 12; As the proposal suggests that the cohort of older people would have an age-range of 60-75 years. How would the common comorbidities like diabetes and hypertension be controlled? As both of these conditions are known to cause cognitive deficits. Understandably, the exclusion of people with such common comorbidities may make a goal of 40 participants challenging to reach. In such a case, is there any consideration for collecting the relevant information and controlling the effect of such confounders at the back end?

(see paragraph above - response to the editor): In the participants’ section (page 15), information had been shared on a health questionnaire /pre-screening to take place on the phone, with the aim to exclude participants with a history of illness, drug use and other health issues (including endocrine disorders, high blood pressure and medication use), as well as multilingualism. We uploaded this questionnaire at the osf.io platform (currently in French language).

13). Page 12; Would any other relevant neuropsychological tests (e.g., semantic and episodic memory test, verbal fluency, etc.) be administered besides MoCA?

The following neuropsychological tests will also be administered during the 1st session to describe participants’ individual differences: 

● The Similarities (Similitudes) part of the Weschler Adult Intelligence Scale (WAIS-III) test (Axelrod, 2002; Schrimsher, O’Bryant, O’Jile, & Sutker, 2007).

● The Pyramids and Palm Trees Test (PPTT) (version images) (Howard & Patterson, 1992) will be used as a measure of semantic performance. 

● The questionnaire Habitudes de Lecture (Reading Habits) (based on (Wilson, Barnes, & Bennett, 2003) as a measure of cognitive reserve (Stern, 2009).

14). What is the statistical plan for behavioral effects? Also, need to mention statistical tests that would be used to probe the relation between behavioral and neural findings. Though the authors say that an exploratory analysis would be performed to investigate that, a little statistical detail would be helpful for the clarity of the planned analyses.

Response times and accuracy rates will be collected for every participant. Participants with a high error rate (outliers) will be excluded from further analysis following data collection to make sure participants do not answer by chance but remain concentrated to the task. Outliers are defined as two standard deviations higher than group average. Trials with errors will be collapsed with correct trials and analyzed in the same single process, given that errors in trials still reflect semantic processing. To account for performance, brain imaging analysis will focus on correct trials only. Behavioral data (RT and accuracy) will be analyzed using mixed- design ANOVA with age as a between-subjects factor and condition (high vs. low demands) as within-subject factor. Accuracy rates will be transformed using Fisher logit approximation to avoid ceiling effects. Group analyses of the imaging data will be performed including behavioral covariates to investigate age group differences in the relationships between brain activity and task performance. Multiple comparisons across the 40 ROIs will be made using false discovery rate adjustments. Analyses will explicitly focus on the relationships between brain activation and task performance. These analyses will identify brain regions where age group differences in activation are dependent or independent of task performance. Time-outs (delayed responses) will be modeled and analyzed separately. Any missing or incomplete data will be excluded (the whole participant).

Reviewer #2: This proposal aims at investigating whether semantic control in younger and older adults follows the pattern predicted by the CRUNCH model. Participants will be performing a semantic decision task whereby they should pick one of two words that they think is more closely related to a target word. The semantic distance between the two choice words and the target word will either be great or small, producing “easy” and “difficult” conditions, respectively. The prediction is that in the easy condition, older adults will exhibit compensatory brain activity with maintained behavioral performance. However, in the “difficult” condition, brain activity will decline and behavioral performance will suffer.

I think this research has good potential, and I’m generally in favor of supporting this project. However, I also do have some concerns and recommendations that should be taken care of before the main project is initiated. I outline these below:

1. Theoretical rigor: Although the CRUNCH models provides a reasonable explanation for compensatory brain activity at low levels of task difficulty, its explanation for high levels of task difficulty is not very convincing (at least to me). Why should brain activity decline for difficult tasks? Why not just plateau? If an increase in brain activity is supposed to compensate for weakened cognitive ability, one would expect brain activity to increase up to a ceiling level and stay at that level. Behavioral performance cannot further improve when the ceiling brain activity is reached. One alternative possibility for why brain activity (and behavior) declines for difficult tasks is that processing becomes “good-enough” or “shallow” to save processing resources (e.g., Ferreira et al., 2002; Ferreira & Patson, 2007; Karimi & Ferreira, 2016). In other words, at high levels of task difficulty, older adults may resort to strategic performance. Such strategic processing may lead to a decrease in brain activity because participants may essentially bypass some aspects of the task. This is more likely for older adults because they don’t have the neural reserve to attend to all aspects of the task. In the current proposal, this may translate into making a quick decision based on whatever information becomes available first, rather than evaluating all the semantic features of the word before making a decision. 

The authors agree that the CRUNCH framework as expressed is interesting and original in its conception. The phenomenon was observed by (Reuter-Lorenz & Cappell, 2008) with a working memory task whereby task demands refers to memory load (the number of items that need to be remembered). Additionally, this phenomenon is referred to by (Cabeza, 2002; Cappell et al., 2010; Fabiani, 2012; Rypma et al., 2007; Schneider-Garces et al., 2010). There is evidence to support the observed increases in activation, plateau and consequent decreases in activation. For example, (Reuter-Lorenz & Cappell, 2008) state: ‘According to CRUNCH, processing inefficiencies cause the aging brain to recruit more neural resources to achieve computational output equivalent to that of a younger brain. The resulting compensatory activation is effective at lower levels of task demand, but as demand increases, a resource ceiling is reached, leading to insufficient processing and age-related decrements for harder tasks’. Also, the authors (Reuter-Lorenz & Park, 2010) state: ‘Therefore, older adults are likely to show overactivation, including frontal or bilateral recruitment, at lower levels of cognitive demand where younger adults show more focal activations. However, as load increases, younger adults may shift to an overactive or bilateral pattern to address task demands, whereas older adults, who may have already maxed out their neural resources at the lower load, show underactivation and performance decline. The predictions of the CRUNCH model have been upheld in several studies of working memory (Cappell et al., 2010; Mattay et al., 2006; Schneider-Garces et al., 2010). This model meshes well with the notion of “reserve” in that individuals with more reserve may reach their resource limit at higher load levels (Stern 2009), making overactivation less likely at lower loads (e.g., Nagel et al., 2009; Smith et al., 2001; Rypma et al., 2007). This may be why longitudinal work has found that overactivation predicted subsequent cognitive decline (Persson et al., 2006). Additional evidence on the increase in activation, plateau and consequent decrease and decline in performance is shared by (Cabeza et al., 2018). We aim to test the CRUNCH model predictions in the field of semantic memory. 

I recommend that the authors go beyond CRUNCH to explain the expected pattern of results, especially for the “difficult” condition.

(See paragraph-response to editor) 

2. Confirmatory data via eye-tracking: This is just a suggestion and I don’t know if the authors have an MRI-compatible eye-tracker. But just in case they do, it would be nice to show that participants fixate on the two choice words more often (by going back and forth between them more often) in the difficult relative to the easy condition. Then, if older adults do any good-enough/shallow processing, their eye-movements should reveal this; they should look at the choice words less often in the difficult relative to the easy condition, but younger adults may show the opposite pattern.

Thank you for your recommendation, which would add enormous value to the task. Unfortunately, this methodology is not supported currently at the neuroimaging unit where MRI acquisitions will take place. However, the authors will keep it in mind as a powerful tool for future studies.

3. Apparent vs. actual difficulty: It is not currently clear how “apparent” and “actual” difficulty may affect the results. If eye-tracking is not possible, the authors can explicitly ask the participants how difficult the current item was after each trial. This way, they can run another analysis on the “participant-rated” easy and difficult conditions and see if the results differ.

(See paragraph above- response to editor) 

4. Neuropsychological tests: Currently, the neuropsychological tests are very limited. How can we ensure that the results are not caused by cognitive abilities other than semantic control, including inhibition, attention, speed of processing, working memory capacity, language knowledge etc.? I recommend that that author collect data on these individual differences and analyze the data accordingly.

The following neuropsychological tests will also be administered in the 1st session to describe participants’ individual differences:

● The Similarities (Similitudes) part of the Weschler Adult Intelligence Scale (WAIS-III) test (Axelrod, 2002; Schrimsher et al., 2007).

● The Pyramids and Palm Trees Test (PPTT) (version images) (Howard & Patterson, 1992) will be used as a measure of semantic performance. 

● The questionnaire Habitudes de Lecture (Reading Habits) (based on (Wilson et al., 2003) as a measure of cognitive reserve (Stern, 2009).

5. Feature vs. association: It is currently unclear if the semantic distance between words is affected by semantic features or semantic associations. Because associations may give rise to stronger automatic cognitive processes such as semantic priming, and because such automatic processing are largely preserved in older adults, I recommend the author try to minimize strongly-associated words and focus more on shared semantic features to increase power.

Indeed, sharing semantic features and belonging to the same taxonomy (e.g. taxonomic relations) vs. being in a semantic association (e.g. thematic relations) may affect semantic processing. We have aimed to construct stimuli that is controlled for as much as possible for several factors given time constraints but also availability of databases. For example, we have matched stimuli for frequency, imageability and length, according to databases mentioned in the manuscript. Also, we have matched stimuli for taxonomic (sharing semantic features) and thematic (semantic associations) relations (see stimuli section). In regards to semantic priming, indeed it has been shown to be spared by aging (Allen et al., 1993; Laver, 2009; Lustig & Buckner, 2004). Given however that semantic priming refers to short stimulus onset asynchrony (eg. SOA < 400ms) (Sass, Krach, Sachs, & Kircher, 2009) whereas our task uses a longer stimulus onset asynchrony (approximately 3.5s), we believe that this is not going to be a confounding factor in our study.

Reviewer #3: This study investigates the influences of task demands on the brain activity serving semantic memory, based on the CRUNCH model. To conduct the study, the authors plan to manipulate the levels of task demands and compare the corresponding differences in neurofunctional activation in young and old age groups. Several expectations are made by the authors: 1) the activation in semantic control regions will be significantly different between young and old participant groups. Specifically, in the low-demand condition, older adults will present significant increases in activation in left-hemisphere control regions compared to the younger participants, reflecting the compensation for higher perceived task difficulty. 2) In the high-demand condition, younger adults will present higher activation in the semantic control regions than older adults, reflecting the compensational overactivation in the older group has reached a threshold beyond which additional neural resources do not suffice. 3) the behavioral performance will also follow the inverted U-shaped relation between compensation and task demands.

Comments

The motivation of this study is clear and the relevant background literature is covered at an appropriate level of detail. It has potential to make some contribution to the understanding of the neural basis of cognitive aging in semantics. However, this paper is difficult to follow in places and the logic and rationale for the authors’ hypotheses are unclear, so I feel some clarification should be made. Additionally, I have a number of concerns about the details of the experimental design and data analysis which need to be better articulated to understand what the authors are proposing.

First of all, the logic and rationale for the authors’ three hypotheses are unclear (Page 9). Based on the CRUNCH model, semantic control regions increase their activations to compensate for increased task demands, whereby after a certain difficulty threshold, available neural resources have reached their maximum capacity and further demand increases lead to reduced activations. However, it’s not clear how they’ve translated this verbally described theory into experimental predictions in hypotheses 1 to 3. Why do they believe, for example, that their high-demand condition will catch older participants at the descending part of the CRUNCH curve and not, for example, on the ascent or at the plateau? It’s not clear to me how they can be confident about this. 

(See paragraph-response to editor): Several papers in the literature have referred to the inverted U-shape relation between task demands and fMRI activation (Rypma 2007, Cabeza et al., 2018; Reuter-Lorenz & Cappell, 2008), and this has inspired our research aims. However, our aim is not to test the validity of the CRUNCH model or the validity of the inverted U-shape itself. This would indeed require more task demand level points. Semantic memory on the other hand is context-dependent and creating task demand levels for semantic memory would be complex. Instead, we aim to test the predictions of the CRUNCH model regarding differences between age groups. These are expressed as: 1) age related over-activation at low loads 2) age-related under-activation at high loads (Cappell et al., 2010). These predictions can be evaluated with 2 levels of task demands only. Our second focus is to test these predictions as manifested specifically in the semantic control network (ROIs).

More specifically, the CRUNCH model was conceived on a working memory task of a quantifiable nature (number of items to retain) (Cappell et al., 2010). Studies that have tested CRUNCH with more than 2 difficulty levels, have done so on working memory ((Fabiani, 2012; Rypma, Eldreth, & Rebbechi, 2007; Schneider-Garces et al., 2010), including recently an evaluation of the model with 4 demand levels on working memory (Jamadar, 2020). Our focus is on semantic memory which is of a different nature, more context-dependent and more difficult to quantify or manipulate for task demands (see for example Patterson et al. 2007 on the salience of semantic features and Badre et al. 2002 on the variability of strength between semantic representations). There is no previous study in our knowledge which has evaluated the CRUNCH model in the context of semantic memory with more than two levels of task demands (see however the behavioral study of Fu et al. 2017). Studies on semantic memory that have examined the impact of differing task demands, have done so with 2 levels (Chiou et al., 2018; Davey et al., 2015; Kennedy et al., 2015; Noppeney & Price, 2004; Sabsevitz et al., 2005; Zhuang et al., 2016). 

Testing the validity of the CRUNCH inverted-U shape on semantic memory with 4 levels of task demands would require a very complex methodology. For example, Jamadar (2020) states: ‘the CRUNCH model is challenging to test because it must be possible to manipulate task difficulty, parametrically, across 3–4 levels. Not all cognitive constructs/tasks are amenable to these requirements’. The stimuli that was developed for the current study (360 words overall) was developed following numerous iterations, evaluations from pilots and peer-reviewed by a team of linguists. Words have numerous connotations and can elicit varying semantic representations. Factors such as frequency, imageability, length, age of acquisition, familiarity, emotional valence, concreteness (among many other factors) interplay with task demands and can thus influence processing of these representations (see for example, Sabsevitz, Binder 2011, Meteyard 2012, Barsalou 2008).

In regards to our stimuli and the required difficulty level to provoke a differential activation between the 2 task demand levels, 1) by design, there is an added level of complexity, provided within the definition of conditions (low-high) and 2) our behavioral data from pilots with 28 participants provide evidence of a significant main effect of task for both accuracy and RTs. Additionally, to account for individual differences of difficulty between participants, we have included in the manuscript: ‘Subsequent analyses will explore this question with heterogeneous slopes models using individualized rescaled levels of task difficulty and will compare brain activation with performance, brain activation with perceived difficulty and performance with perceived difficulty. using a throughput measure (Schneider-Garces et al., 2010). This approach will determine how the relationship between individual task difficulty and brain activity is affected by age group’. 

As such, our goal is to test the main predictions of the CRUNCH framework as first evidence of its applicability to semantic memory, manifested with differential activation in the semantic control network. We have rephrased these sentences in our hypotheses to better explain that we focus on the predictions rather than the validity of the whole CRUNCH model. 

Do they, for example, have some behavioral data indicating that older people are at ceiling in the low-demand condition but are more error-prone in the high-demand condition? This would help to reassure me that older people will have exceeded their processing limits in the high-demand condition, particularly if young people showed a different pattern.

(See paragraph-response to editor about behavioral data).

The concept of “perceived task difficulty” in different age groups is introduced to help establish the authors’ hypotheses, but the relation between “perceived difficulty” and actual task demand/difficulty and “perceived difficulty” fits in the CRUNCH theory are not well discussed. Additionally, the authors assume that by manipulating the two levels of task demands (low and high) they can precisely locate the “perceived difficulties” in the increasing and descending part of the inverted U-shaped model, I feel this is not that easy to achieve.

(See paragraph/response to editor about actual vs. perceived difficulty).

I also found the methods of data analysis unsatisfactory and lacking in detail.

First, hypotheses seem under-specified:

Hypothesis 1 predicts a demand x age interaction in accuracy and RT and in BOLD activation in 6 separate ROIs (pg. 9). These requires 8 statistical tests. To conclude that the hypothesis is upheld, do all of these effects need to be significant? Or only some of them? If so, which ones? Confusingly, in the data analysis section it’s stated that this hypothesis will be tested at the whole brain level, contradicting the description provided earlier in the paper.

Thank you for your comments. The ‘Defining the anatomical/functional ROIs’ section was rewritten. The hypotheses will be tested in 40 ROIs identified from a recent comprehensive meta-analysis of semantic control (Jackson 2021). To conclude that a hypothesis is upheld, we expect at least one region of interest (ROI) to demonstrate significant activation. We will use false discovery rate correction for multiple comparisons across the ROIs as described in the manuscript (section: Defining the anatomical/functional ROIs). The manuscript has been edited to clarify and expanded to address these important points. In addition, many other details have been added throughout the methods section to improve clarification and replication based on the excellent reviews we have received. 

Hypothesis 2 predicts that in the low-demand condition, participants will make minimal errors. How will this be tested? What is the definition of minimal errors? It is also hypothesized the older people will present longer RTs and more activation in control regions. Again, how many tests have to be significant to accept this hypothesis?

Indeed the term ‘minimal errors’ is confusing. We replaced this with ‘both young and older participants will perform equally in terms of accuracy and with less errors than in the high-demand condition’. To support hypothesis 2, at least one brain region needs to show significant activation. 

Hypothesis 3 predicts that in the high-demand condition, only young adults will perform optimally (again, defined how?), that they will perform better than older people in accuracy and RT, that older people will show less activation in left-hemisphere semantic control regions (all 6? Or is just 1 enough?) and more activation in right-hemisphere homologues (again, 6 tests here?).

The term optimally was replaced with ‘better (than older adults)’. In regards to brain activation, we expect that at least one ROI will show a significant decrease in left-hemisphere ROIs and at least one ROI will show a significant increase in right-hemisphere ROI. We are interested to see if the same regions will be activated for all 3 hypotheses. This would demonstrate which regions activate consistently as part of the ROIs and are thus important for semantic control.

Second, critical information is lacking about the ROIs. There are 12 ROIs in total in this study (Page 22, 23), but very little information is provided about how they are defined and how they will treated statistically. The Neuromorphometrics atlas that the authors want to use to define the ROIs does not include some of the specific regions they discuss. For example, this atlas does not have a single IFG area. How will the various IFG subregions in this atlas be combined to make a single ROI? It does not have an area labelled dorsal inferior parietal cortex. How will this be defined? It does not divide the MTG into anterior and posterior parts, so how will pMTG be defined? And so on.

The description of the region of interest definition has been completely rewritten. We will use the 20 identified locations (40, bilaterally) resulting from a recent meta-analysis on semantic control tasks combining 126 comparisons and 925 brain activation peak locations (Jackson 2021). This is the most up to date and comprehensive study of semantic control. Using these 40 locations as centers, spheres of activation will be extracted from our data to serve as ROI data, following the recent protocol for a similar study of aging and the CRUNCH model (Jamadar 2020).

In addition, as stated earlier, the same hypotheses are being tested in multiple ROIs. The authors should state how they are handling issues of multiple comparisons as some sort of correction seems necessary here.

There will be forty ROIs and the false discovery rate method will be used to correct for multiple comparisons (see section ‘Defining the anatomical/functional ROIs’).

Finally, more information should be given on the discussion of power calculations (Page 11, 12). The authors mention that “power estimates calculated that a sample of 38 participants per age group would be sufficient to detect age group differences bilaterally in the IFG, AG, dACC, dIPC, and large areas of the PFC”. What are the expected effect sizes in each of these regions, what alpha level has been used to estimate power (and does it account for multiple comparisons?) and what power is achieved with this sample size?

The power analysis section was rewritten (p.32). The expected effect size is 90% and alpha level is 0.05. We uploaded the table of effect sizes of the Ferre et al. (2020) used as input for our power analysis at osf.io. 

Minor comments

Page 9. The sentences of the three hypotheses are long and difficult to understand. For example, the paper says: “If this interaction is not found between task demands and age, it is expected that … (Line 190)”. The causal relationship between hypothesis 1 and 2/3 that the authors are claiming here is not clear.

This was addressed in the manuscript.

Page 16. The estimation of the duration of the fMRI experiment does not add up. There are two runs in total in this experiment, and the duration for each run is 9:45 minutes. The authors suggest that the fMRI session will take 90 minutes, which is much longer than I expected even considering all the related preparation procedures.

Indeed, the actual scanning session will last approximately 20min. We estimate a generous 90min for the whole participant appointment, assuming extra time for participant’s eventual delay, preparation (e.g. receiving instructions and practicing with practice trials in a different room, walking to the MRI room, getting dressed adequately, putting on MRI-compatible glasses if the participant wears, pregnancy tests if necessary etc). Especially given the COVID-19 situation, extra time is required to disinfect the areas as needed and since not more than 2 participants are allowed in the testing room. The participant is not obliged to stay for 90min if the session is finished in less time. This was clarified in the manuscript.

Reviewer #4: Haitas et al. present a protocol for a registered report for an fMRI experiment testing the CRUNCH model of cognitive compensation. They propose to compare 40 young and 40 older adults on a semantic judgement task. In general, the experiment is well described, and the manuscript is clearly written. I am a little concerned that perhaps the design requires some additional consideration, and I outline this below. I welcome this registered report examining the CRUNCH model as numerous studies in this area are under-powered and poorly designed; as such I believe this work may have a substantial impact in this area.

MAJOR COMMENTS

1. I am surprised that the experimental design only seems to include 2 levels of task difficulty, when the literature states that 3-4 levels of difficulty (Cappell et al.; Reuter-Lorenz & Cappell; Fabiani et al.) is required to determine the hypothesised non-linear relationship between task difficulty and brain activity. The authors refer to this non-linear relationship in the introduction (‘inverted U-shaped one’, page 4). How can this experimental design adequately test the CRUNCH model with only 2 levels of difficulty? I would recommend adding at least a third level of difficulty to the experimental design.

(see paragraph above-response to editor)

2. A few points about the fMRI data analysis:

a. it is becoming much more common to include derivatives of the motion parameters in the GLM as well as the motion parameters themselves.

Thank you for your comments. 

This was corrected in the manuscript ‘fMRI data analysis’ section (p.44): For the 1st level (intrasubject) analysis, a General Linear Model (GLM) employing the canonical Hemodynamic Response Function (HRF) and its derivative both convolved with a model of the trials will be used to estimate BOLD activation for every subject as a function of condition for the fMRI task. The inclusion of the derivative term accounts for inter-individual variations in the shape of the hemodynamic response. Each participant’s fMRI time series (2 runs) will be analyzed in separate design matrices using a voxel-wise GLM (first-level models). Movement parameters obtained during preprocessing, and their first and second derivatives, will be included as covariates (regressors) of no interest to reduce the residual variance and the probability of movement-related artifacts. 

b. Also, is there a reason why HDR derivatives are not included in the model? Older participants may have different HDR shape/timing compared to younger people.

(See paragraph above-response to the editor)

c. I also strongly suggest accounting for confounds in the model – at the very least, accounting for grey matter/cortical thickness differences between the groups, if not other physiological parameters like cardiovascular parameters, sex, haemoglobin, etc. Differences in grey matter volume between the groups should be explicitly tested for and controlled in the analysis.

We agree and will aim to control for sex (male=female) and will be used as a covariate in all analyses. Controlling for global cortical thickness or volume would be ideal, but will be problematic. There are expected to be large age-related grey matter volume differences. Therefore, the introduction of a highly collinear covariate into the model will decrease the statistical power to detect age-related differences. First-level fMRI statistical analyses will include motion parameters and their derivatives to account for additional variance in the data.

d. Also, is the ‘baseline’ implicit baseline or the control task? 

It is a control condition, and we corrected for this in the manuscript. 

Is the post-hoc tests of age group (pg 22 ln 469) a t-test? 

Yes, it is a t-test and this was corrected in the manuscript.

How does this control for potential cardiovascular confounds between the age-groups?

Though some studies have found that differences in BOLD activity may be attributed to cardiovascular differences between young and old (Tsvetanov et al. 2015), other studies have shown that neurovascular coupling does not significantly change with age (Grinband et al. 2017, Kannurpatti et al. 2010). The inclusion of the HRF derivative will account for inter-participant and inter-group variations in the hemodynamic response to stimuli. It is also reported that comparing differences between individuals accounts for potential cardiovascular differences (d’Esposito et al. 2003).

e. I am a little confused about whole brain vs. ROI analysis. Most of the analysis section seems to be referring to a whole brain approach but then the hypotheses are focused on the ROIs. Which parameters will be calculated from the ROIs? How will they be analysed?

The description of the region of interest definition has been completely rewritten (see ‘Defining the anatomical/functional ROIs’ section p51). We will use the 20 identified locations (40 bilaterally) resulting from a recent meta-analysis on semantic control tasks combining 126 comparisons and 925 brain activation peak locations (Jackson 2021). This is the most up to date and comprehensive study of semantic control. Using these 40 locations as centers, spheres of activation will be extracted from our data to serve as ROI data, following the recent protocol for a similar study of aging and the CRUNCH model (Jamadar 2020). We will therefore utilize the MarsBar (Brett et al. 2002) tool to extract data from spheres of a radii of 10mm around the 40 locations identified in the metaanalysis for analyses. 

f. I think that if the analysis approach is modified (e.g., including derivatives of HDR & motion parameters, grey matter, etc.) then the power calculation should be revisited.

The data we used for our power analysis (Ferre 2020) already included motion parameters in their statistical models: ‘A motion-censoring procedure was applied to remove unwanted motion, physiological and other artifactual effects from the BOLD signal. An ART-based functional outlier detection method was used, as implemented in the CONN toolbox (Mazaika, Whitfield & Cooper 2005). The threshold was established using the maximum voxel displacement with a scrubbing criterion established at 0.9 mm scan-to-scan head motion or global signal intensity of 5 SD above the mean signal for the session (Mazaika, Hoeft, Glover & Reisse 2009, Whitfield-Gabrieli & Nieto-Castanon 2012).’ The task of Ferre et al. (2020) consisted of a block design (12 blocks lasting 17.5s). Due to their use of a block design, inclusion of HDR derivatives in their designs offers little benefits and was not done. Our power analysis based on their work has been improved and rewritten. Based on their results the current study will have sufficient power. 

3. One of the reasons why I welcome this registered report is the recent study by Jamadar et al. in Neuropsychologica. These authors suggest that there may be publication bias in the reporting of the CRUNCH effect, and perhaps the effect is not as robust as believed. It would be worthwhile mentioning this criticism in the manuscript, maybe in the introduction.

Thank you for sharing this very reference, very important for the CRUNCH model and the reproducibility of science in general. We added it in our manuscript. 

4. “Power estimates calculated that a sample of 38 participants per group would be sufficient to detect age group differences bilaterally …” does this account for multiple comparisons correction? Could the authors provide an alpha level here, and note how they will correct for multiple comparisons across the 10 (?) ROIs. 

In regards to power analysis, the expected effect size was 90% and alpha level were 0.05. Correction for multiple comparisons will use the false discovery rate across the 40 ROIs (Benjamini and Hochberg 1995). 

Also, what does ‘large areas of the PFC’ mean? Is this a large ROI across subsections of the PFC?

This part was deleted/rewritten.

5. Stimuli description – I didn’t quite follow this section. Have the authors already run a pilot in 100 people to collect the semantic relationship data? Are these people similarly aged as the experimental groups? I could imagine that a semantic relationship might exist for one age group and not another (e.g., stream – music), have the authors assessed if the semantic relationships are maintained across age groups? 

The 100 people mentioned here refers to people who participated in the survey conducted for the ‘Dictionnaire des associations verbales (sémantiques) du français’ (http://dictaverf.nsu.ru/dict). This is an online database whereby respondents provided the first item that came to their mind when provided with a given target. This database was accessed and used to construct our stimuli, but is not part of the current project. There is no information available on the age of respondents of this online dictionary. To our knowledge, there is no database available in French that has controlled for words’ association according to age groups. We have controlled our stimuli for frequency and imageability, thus making it equally ‘accessible’ and understandable to both age groups. Pilot evaluations of the stimuli were conducted by both younger and older adults and any consequent changes were made (e.g. for unpopular words).

How many participants were excluded on the basis of the correlation < 0.6 threshold?

In regards to participants who were excluded on the basis of the <0.6 threshold, these participants did not participate in our stimuli pilots, but were participants who evaluated imageability of word stimuli in order to give these items a score (since these word stimuli were not included in the DesRochers imageability database). As such, 31 participants (age range 23-74) were requested to score 307 words for imageability in a scale from 1 (very low) to 7 (very high). Among these 307 words, 30 were ‘test’ words (already had a score in the imageability database to be able to compare and correlate their scoring) whereas the remaining ones did not (they were the actual word stimuli of our interest). We used Pearson’s correlation to correlate the 2 scores provided by the participants. We excluded 6 participants for giving a score to the 30 test words which had a correlation value of less than 0.6 from the one available in the database, as it was deemed they were not concentrated on the task

Is there behavioural data to confirm that ‘hard’ trials are actually difficult, and ‘easy’ trials are actually easy – and comparable between the age groups? One of the challenges of the CRUNCH model is the definition of ‘difficulty’ is not as clear cut in all cognitive domains as it is in the memory domain, and I think that it is worthwhile confirming that task difficulty is working as expected, and similarly across age groups.

(See paragraph-response to editor on behavioural data).

MINOR COMMENTS

1. A figure would be useful to illustrate the hypotheses.

Three figures were added to illustrate our hypotheses on brain activation, accuracy and RTs differences between young and older across different levels of task difficulty (see page 13).

2. The motion exclusion strategy is fairly conservative. Are the parameters given (2mm/1deg) acute motion or drift across time? Perhaps there may be more data loss for the older than younger participants with this criterion.

It concerns acute motion. The sentence now reads “Participants with estimated acute motion parameters of more than 2mm, or 1-degree rotation, between scans in any direction, will be excluded.”

3. The TR is quite long – is there a reason for this? 

Yes, the TR is relatively long. This duration is used because it maximizes the signal to noise ratio for our scanner and allows for full brain coverage of 150mm in the z-direction and a voxel size of 2.5x2.5mm in-plane resolution. 

And is there a reason why the EPI resolution is not isotropic?

The relatively long TR and isotropic voxels are specifically chosen to correct for the non-homogeneity of gradients on the temporal regions and to minimize signal loss in these regions.

4. Participants respond with ‘any of their fingers’ – it is typical to standardise the response keys across subjects.

That sentence now reads: “Participants will select their responses using the index fingers of both hands on the MRI-compatible response box. A response on the right will be with their right hand and a response on the left with their left hand.”

5. I think that the activation height threshold noted on pg 22 ln 478 needs ‘uncorrected’ specified.

This part was deleted, and the’ fMRI data analysis’ section was rewritten.

6. “Participants who qualify for the fMRI scanning session…” do participants need to meet a certain threshold of performance in the 15 practice triads to qualify? Does ‘qualify’ here mainly mean ‘meet inclusion criteria’? This is unclear. 

Qualify for fMRI scanning session refers to the inclusion/exclusion criteria presented in section ‘Participants’, namely who meet the inclusion criteria from health questionnaire, MRI-compatibility questionnaire, MOCA and Edinburgh Handedness Inventory scale. There will be no performance threshold to achieve from the 15 practice triads. 

Also will session 2 always be one week later, or will there be a window of opportunity.

The 2nd session will take place one week later (maximum within 2 weeks), depending on participant availability. 

7. PLOS authors have the option to publish the peer review history of their article (what does this mean?). If published, this will include your full peer review and any attached files.

Do you want your identity to be public for this peer review? For information about this choice, including consent withdrawal, please see our Privacy Policy.

Reviewer #1: No

Reviewer #2: Yes: Hossein Karimi

Reviewer #3: No

Reviewer #4: No

Reviewer #5

It was a pleasure to review this fMRI registered report manuscript. I like the study idea and general design, and the authors’ commitment to COBIDAS guidelines, power analysis and preregistration. And this is a good study to do to test CRUNCH. The operationalisation of (objective) task difficulty is also meaningful and well-motivated. So there are many positives – but there are some quite deep theoretical issues in this literature on ageing and possible compensation that should be taken into account in the Introduction and methods, mainly by more explicitly considering alternative hypotheses, and by more clearly defining key terms and the statistical tests that are needed to support interpretations.

Below are my main points in chronological order through the manuscript. Points 2, 3 (particularly a-c) and 7 are the most important. More minor issues are listed at the end.

1. In the first part of Introduction, the explanatory emphasis is on age-related preservation (eg., line xx ) or even improvement (eg line 35) of semantic cognition with age but there is only a bare mention of semantic control, the focus of this study. This is given a good introduction later, but it would help the reader unfamiliar with the concept, or the existing studies, to preview this right at the start.

Thank you for your thorough review and constructive comments. 

We addressed this issue in the introduction, adding the phrase: ‘The relative preservation of semantic memory performance in older adults when compared with other cognitive fields (Hoffman & Morcom, 2018; Salthouse, 2004, Fabiani 2018) could be partly justified by the proposed dual nature of the semantic memory system, as expressed within the controlled semantic cognition framework (Jefferies 2013, Lambon Ralph et al. 2017, Jackson 2021, Hoffman 2018)’. 

2. The CRUNCH theory is complex (as you imply) and is probably interpreted in more than one way, with potentially critical differences in operationalisation at least. The Introduction could be clarified in the following ways:

a. A figure of the inverted U to explain the predictions for this task would be really helpful.

As we will not be testing the inverted-U shape relation between task demands and fMRI activation (see response to editor above), but we will be testing instead the CRUNCH predictions, we included 3 figures related to the hypotheses, namely on the relation between task demands with: RTs, accuracy and activation.

b. Please specify what is meant by ‘urgent’ task requirements – if you don’t find this convincing, you might use examples from the original papers.

This was further elaborated in the manuscript in the paragraph mentioned, but also in an added section on alternative to CRUNCH explanations.

c. In introducing evidence for CRUNCH, please specify explicitly the pattern/s results referred to as ‘CRUNCH-like overactivations’.

We elaborated this paragraph with more precise details and renaming CRUNCH-like as ‘Compatible with CRUNCH, increased overactivations with maintained performance...’.

d. Is it your interpretation of the CRUNCH theory that CRUNCH-like patterns of brain activity will be found for tasks tapping cognitive functions that show age effects, but not for tasks that do not? this is never explicitly stated but is implicit in the different predictions for the baseline task.

We hope we understood your comment well. The CRUNCH framework is thought to apply not only to older but to young adults as well, who would demonstrate increased compensatory activations once task demands exceed a certain level. The defining feature of the CRUNCH model is thus task demands whereas predictions are made within-person as a function of task demands and availability of neural resources (Festini 2018). Similarly, ‘individuals with a neurological disease or disorder may compensate for their disorder-related deficits in ways similar to those described here for healthy older adults’ (Cabeza et al., 2018). Semantic memory of younger adults is affected by increasing task demands (Badre, Poldrack, Paré-Blagoev, Insler, & Wagner, 2005; Krieger-Redwood, Teige, Davey, Hymers, & Jefferies, 2015; Noonan, Jefferies, Visser, & Lambon Ralph, 2013). As load increases, younger adults may shift to an overactive or bilateral pattern to address task demands, whereas older adults, who may have already maxed out their neural resources at the lower load, show underactivation and performance decline (Reuter-Lorenz & Park, 2010). The manuscript mentions: ‘Task effects within each age group will be tested and activation is expected to be of greater amplitude in the high vs. low condition in both young and old age groups’.

e. The 2008 CRUNCH paper specified that compensatory overactivation might compensate *for* either a region’s own declining efficiency, or for deficiencies elsewhere. Which is the case for semantic control? It seems to me that you might expect additional activations in older people in task-general (executive control as opposed to semantic control) regions. It might be worth testing for this and asking whether these age differences are also more pronounced in the harder condition (see Hoffman and Morcom 2018 for related observations).

Indeed, during the more high-demand condition of the semantic task, increased activations could also be expected in the task-general or, multiple-demand network and not exclusively in the semantic control network. Regions related to semantic control are thought to be largely overlapped by the neural correlates of the semantic network (Jackson 2021) but also thought to largely overlap with regions related to the ‘multiple-demand’ frontoparietal cognitive control network (Lambon Ralph, Jefferies, Patterson, & Rogers, 2017; Vincent, Kahn, Snyder, Raichle, & Buckner, 2008) and as observed in (Hoffman & Morcom, 2018). Language and executive functions are overall intertwined given that regions such as the left inferior frontal gyrus and the PFC are proposed to serve both executive and language functions (Diaz, Rizio, & Zhuang, 2016). Given the complexity of the issue and the sometimes conflicting findings, we decided to focus on the stated ROIs with followup exploratory analyses that will explore how differential brain activation is related to task performance. 

3. The CRUNCH theory deliberately combines a set of observations about activation patterns with the assumption that they are compensatory. This is absolutely fine for a theory, but does not mean that the theory is the only one that could explain such activation patterns in relation to load. I suspect that you are testing here for a CRUNCH activation pattern but not the compensatory interpretation. Or, perhaps there is a way in which the proposed study can adjudicate between compensatory and non-compensatory theories. Or, are there different forms that CRUNCH might take?

a. The concept of compensation is muddy in the literature, therefore needs explicit definition in this context. Different definitions are used in the literature, or none. For example, the results of our (Hoffman & Morcom 2018) meta-analysis are described (line 119) as showing ‘CRUNCH-like compensatory increases in activation’. Assuming that you mean that the additional activation is helping participants to perform better than they would have done without it, this was not our sole interpretation, but one possible explanation (see next point). Moreover, without data from manipulations of load, CRUNCH was not directly tested in that meta-analysis (which is why your study is needed).

Indeed, determining if neurofunctional differences between young and older adults have a purpose or are the result of aging, is complex. The following definition for compensation was added in the manuscript: ‘the cognition-enhancing recruitment of neural resources in response to relatively high cognitive demand’ (Cabeza et al. 2018). We are sorry for misinterpreting your metaanalysis. We rectified this by replacing at this point (and the whole of the manuscript) the term CRUNCH-like with compensatory. Also, we rephrased the sentence as ‘...the authors reported increases in activation in semantic control regions in older adults …’. Throughout the manuscript we refer to the de-differentiation account more explicitly as an equally valid interpretation. 

b. Without a definition of compensation and consideration of alternative accounts, the following statement is currently not supported (line 128):” Evidence therefore exists that a significant increase in activation of semantic control regions may compensate for increased task demands and favor semantic memory performance in both old and younger adults”.

We corrected this inaccuracy by replacing the term with: ‘Evidence therefore exists for a correlation between significant increase in activation of semantic control regions when faced with increased task demands, which could be indicative of the compensation account, favoring semantic memory performance in both young and older adults’.

c. An alternative set of theories assume that brain activation becomes less specific or efficient with age (eg Park et al., 2004). Please consider these in your interpretations of the literature and predictions for the proposed study. for example, nonspecific responses might follow a non-compensatory CRUNCH-type pattern in that the overactivation would track difficulty and interact with age, but would not actually help people perform better at the task (compared to how they would perform without the overactivation).

We elaborated more on the de-differentiation account in the introduction.

d. Another relevant theory is that of cognitive flexibility, closer to but distinct from other ideas about compensation (see Lovden et al. 2010) – the proposal that overactivation with age reflects greater engagement of task-general processes (see also 2e above).

We elaborated more on the plasticity account in the introduction.

e. A related point: you refer to ‘attempts at compensation’ (line 153, re. Meinzer et al. 2009). Would there be any evidence that could convince you that nonspecific activity is just that, rather than an attempt at compensation? It seems to me that the answer is no, and therefore this is not a helpful term. If I’m wrong, please specify how this could be tested meaningfully.

The ‘attempt at compensation’ was removed and will not be tested. Distinguishing between the de-differentiation accounts and the attempted (but failed or partly) compensation as discussed by Cabeza et al. (2009), would be complicated, since both would present overactivations with reduced performance.

f. When referring to literature that might support CRUNCH, please specify how it’s established that brain activity is ‘beneficial’ or ‘detrimental’ to performance – e.g. in the Meinzer et al study referred to in the last point. There are different ways of trying to do this, none very satisfactory in cross-sectional brain imaging studies (see Morcom and Johnson, 2015; no need to cite us, we also cite others who have made similar points in this synthesis & review).

Thank you for pointing out this very relevant article and critique. We further elaborated on the methodologies used in the cited literature. 

4. I’m concerned about the assumption that CRUNCH relates to ‘perceived’ task difficulty (lines 115,166).

a. Is there anything to support the assertion that perceived task difficulty tracks *between group* as opposed to within-participants increases in response times? I would strongly recommend restricting your CRUNCH predictions so they relate to difficulty that can be measured in some way. If you want to address perceived as opposed to objective difficulty, this should be measured too.

In regards to perceived task difficulty, we added an additional session with participants one week following the fMRI acquisitions, whereby they will rate each triad on a difficulty likert scale 1-7 (1: very easy, 7: very difficult). We will further correlate perceived (participant-rated) difficulty with performance (accuracy and RTs) as well as fMRI activation, to evaluate if the results differ. Rates of perceived difficulty will be collected in a basic descriptive manner for this study whereas no hypotheses will be formulated according to them, as its analysis would deserve its own study and is considered as a future direction. As such, the average perceived difficulty per participant will be used as a measure to compare between groups. The behavioral data we collected demonstrate that there are group differences in task performance.

b. I appreciate that there is an idea to do exploratory analyses of rescaled task difficulty (lines 218 on) but this should either be explained in more detail or omitted as it is too general for preregistration to be meaningful. What do you expect to find that is different and what will it signify?

The relationships between brain and behavior will be done within the ROIs and no brain wide exploratory analyses will be done. Analyses will be a difference in slopes analysis to determine if and where in the brain brain-behavior relationships differ between age groups.

5. Regarding the task, is there any benchmark to decide whether the ‘low’ demand condition is low enough, and the ‘high’ demand condition is high enough? should this be calibrated within Subjects? (this probably relates back to point 4b).

Please see the section on behavioral data from stimuli pilots added in the manuscript. For within-subject calibration based on performance, triads represent low-demand and high-demand conditions by design based on dictaverf for the thematic condition e.g. biggest distance possible between distractor and target for thematic relations based on dictaverf for the low-demand condition, and smallest distance possible based on dictaverf for the high-demand condition. Calibrating at the participant group level would be great, but would require a full other study.

6. Please clarify the predictions about the baseline task (line 209 onwards) – do you mean that you expect no CRUNCH patterns in this task compared to an implicit (fixation?) baseline? If so I agree this is a good control comparison but could be spelt out more clearly.

We added in the manuscript: ‘No CRUNCH effects are expected in the control condition’. For an explanation of the choice of control condition which is a high-level active one vs. a low-level passive one, see (Binney, Hoffman, & Ralph, 2016) and (Sachs 2008) for this type of letter categorization task. In the manuscript it is stated: ‘A control baseline condition was designed to maximize perceptual processing requirements and minimize semantic processing ones (Binney, Hoffman, & Ralph, 2016; Gutchess, Hedden, Ketay, Aron, & Gabrieli, 2010). As a positive control, within group comparisons with the control baseline condition are expected to show activation in the primary visual and motor cortices, which are involved with viewing of the stimuli, response preparedness and motor responses (Geng & Schnur, 2016; Kennedy et al., 2015; Sachs, Weis, Krings, Huber, & Kircher, 2008)’. 

7. It’s great to see a power analysis was done but a lot more detail is needed. I have not used fMRIPower myself but believe it requires pilot data, which are not mentioned here although they are mentioned on the linked OSF site. The procedure should be specified, and it is essential at least to state the criteria entered into the power analysis, usually the required power, the alpha, the effect size, and the contrast and the model that the effect size pertains to. Also, does the Boston naming task show similar age effects to the proposed experimental task?

The power analyses have been redone using the same ROIs as for the proposed study (Ferre et al. 2020) and rewritten in the section Participants (manuscript, p. 32). In regards to power analysis, the expected effect size was 90% and alpha level are 0.05. We uploaded the table of effect sizes of the Ferre et al. (2020) used as input for our power analysis in the osf.io. The Boston Naming Task is a picture naming task measuring word retrieval, whereas our task is based on the word version of the Pyramids and Palm Trees test and measures semantic memory and the ability to access detailed conceptual information to form associations between them. Though the tasks are not perfectly similar, they both access semantic memory and activate broadly similar regions. Our choice was largely guided by the availability -or lack of- statistical maps following our request from authors of studies similar to ours.

8. Behavioural data analysis should include tests for age effects according to difficulty, please add. How does group-level performance on the task relate to the CRUNCH predictions and what do you expect to find? Some are given in the Intro, but they should be summarised in the methods too more formally, as for the fMRI data. Its also not mentioned whether age diffs in performance are expected under low demand?

To account for performance, brain imaging analysis will focus on correct trials only. This important point is now addressed in the manuscript. Analyses are now described which will explicitly focus on the relationships between brain activation and task performance. These analyses will identify brain regions where age group differences in activation are dependent or independent of task performance. Pilot behavior data, included in the manuscript, demonstrate task demand differences in both age groups.

9. fMRI hypothesis tests are for the most part clearly specified. But hypothesis three, regarding age-related activity increases in right semantic control regions, needs a bit more detail. Is the hypothesis of localised additional activation or of a change in lateralisation? If the latter, a laterality comparison would be both more sensitive and more specific to the hypothesis. 

The hypothesis concerns a localized additional activation in the left and right hemispheres with aging.

More minor points

10. Please clarify what is meant by general skills in this sentence on p.3: “Though several age-focused neurofunctional reorganization phenomena (e.g. HAROLD (Cabeza, 2002) and PASA (Davis, Dennis, Daselaar, Fleck, & Cabeza, 2008)) aim to explain how aging affects general cognitive skills”.

HAROLD and PASA claim to be a general aging re-organization phenomenon that are applicable to all cognitive fields. For HAROLD: ‘The model is supported by functional neuroimaging and other evidence in the domains of episodic memory, semantic memory, working memory, perception, and inhibitory control’ (Cabeza 2008), and for PASA: ‘ Taken together, these findings demonstrate the validity, function, and generalizability of PASA, as well as its importance for the cognitive neuroscience of aging (Davis et al. 2008). To be more accurate, the sentence was restated as : ‘..aim to explain how aging affects cognitive skills in general’.

11. Please specify the interaction to be found in the Introduction briefly, ~line 190, as well as in the Methods.

This was addressed in the manuscript.

12. Regarding the task, has this task been time-limited in other studies? This may differentially impact older people. It’s not a problem if this is usual practice but might be if it is not.

The task was developed explicitly for this study, so it has not been tested elsewhere. Data from our pilots were added in the manuscript in the section stimuli description, including how the conditions differ between young and older, as well as between low and high-demand conditions. This pilot data also demonstrated that out of the 4350 trials administered to the 29 participants, 4123 of the trials were responded to within the 4 second response window. Therefore, the timing of this study will capture 95% of the responses. The instructions and training on the task with participants will also stress the need to respond within a four second window. 

13. Thanks for sharing the stimuli. Some further information would be helpful in the eventual online material: i) who were the participants in the imageability task, and what was the procedure, and ii) how many triads were taxonomically and how many thematically related, with any per stimulus metrics.

i) Imageability ratings were collected for words that did not have a value in the DesRochers imageability database. As such, 31 participants (age range 23-74) were requested to score 307 words for imageability in a scale from 1 (very low) to 7 (very high). Among these 307 words, 30 were ‘test’ words (already had a score in the imageability DesRochers database to be able to compare and correlate their scoring) whereas the remaining ones did not (they were the actual word stimuli of our interest). We used Pearson’s correlation to correlate the 2 scores provided by the participants. We excluded 6 participants for giving a score to the 30 test words which had a correlation value of less than 0.6 from the one available in the database, as it was deemed they were not concentrated on the task. ii) Half of the triads are taxonomically and half are thematically related. As such, there are 30 taxonomic triads and 30 thematic triads in the low-demand condition, as well as 30 taxonomic and 30 thematic in the high-demand condition. The triads are available at OSF, we will also add the related metrics.

14. Anyone wanting to replicate this study will also need the eventual stimulus lists with ISIs – these may be in the EPrime files (I don’t have access) but please also share them in an open format like csv.

The intertrial times are now included in a CSV file which has been uploaded to the OSF sharing platform.

15. Why not use BIDS format for the imaging data to maximise shareability?

We will organize our data according to BIDS at the conclusion of the study. The manuscript has been updated to state this.

16. Preprocessing (line 430): do you mean ascending-interleaved rather than ascending (as mentioned in the previous section)?

Yes, we do and this has been fixed in the manuscript.

17. fMRI analysis (line 456) do you mean to specify a high-pass of 200s?

A high pass filter with a temporal cutoff of 200 seconds will be used. This is equivalent to a cutoff of 0.005 Hz. The software SPM specifies their filters with the temporal period.

18. For multiple comparisons correction, I appreciate that you are specifying the updated version of 3dClusterSim but please give a number if you can.

Multiple comparison correction will use the false discovery method across the multiple ROIs used in the study.

---

## [Decision Letter · Decision Letter 1]

29 Mar 2021

Age-preserved semantic memory and the CRUNCH effect manifested as differential semantic control networks: an fMRI study

PONE-D-20-27105R1

Dear Dr. Haitas,

We’re pleased to inform you that your manuscript has been judged scientifically suitable for publication and will be formally accepted for publication once it meets all outstanding technical requirements.

Kind regards,

Anna Manelis, Ph.D.

Academic Editor

PLOS ONE

Additional Editor Comments (optional):

Reviewers' comments:

Reviewer's Responses to Questions

**Comments to the Author**

1. Does the manuscript provide a valid rationale for the proposed study, with clearly identified and justified research questions?

Reviewer #1: Yes

Reviewer #2: Yes

2. Is the protocol technically sound and planned in a manner that will lead to a meaningful outcome and allow testing the stated hypotheses?

Reviewer #1: Yes

Reviewer #2: Yes

3. Is the methodology feasible and described in sufficient detail to allow the work to be replicable?

Reviewer #1: Yes

Reviewer #2: Yes

4. Have the authors described where all data underlying the findings will be made available when the study is complete?

Reviewer #1: Yes

Reviewer #2: Yes

5. Is the manuscript presented in an intelligible fashion and written in standard English?

Reviewer #1: Yes

Reviewer #2: Yes

6. Review Comments to the Author

You may also provide optional suggestions and comments to authors that they might find helpful in planning their study.

Reviewer #1: I don't have any additional suggestions/clarification requests. The authors have satisfactorily addressed all my comments.

Reviewer #2: I read revised research proposal as well as the authors’ response. I think the revised version of the proposal has immensely improved in terms of clarity, comprehensiveness of the relevant theories and experimental rigor. The authors have addressed all of my methodological and theoretical concerns. I therefore recommend an accept decision.

I have to last recommendations for the authors before they launch the study:

1.In relation to semantic distance, it is not clear from the paper if the norms come from younger adult respondents only. (My guess is that they do?). If yes, maybe norming the stimuli with older adults may help gain a better understanding of the degree of semantic relatedness for each group (separately).

2.I would make sure there’s no phonological overlap between the triads. We know that older adults are more susceptible to phonological interference and might therefore be disproportionately affected by such overlap.

7. PLOS authors have the option to publish the peer review history of their article (what does this mean?). If published, this will include your full peer review and any attached files.

Reviewer #1: No

Reviewer #2: **Yes: **Hossein Karimi

---

## [Editor Report · Acceptance letter]

14 May 2021

PONE-D-20-27105R1 

Age-preserved semantic memory and the CRUNCH effect manifested as differential semantic control networks: an fMRI study 

Dear Dr. Haitas:

I'm pleased to inform you that your manuscript has been deemed suitable for publication in PLOS ONE. Congratulations! Your manuscript is now with our production department. 

Kind regards, 

on behalf of

Dr. Anna Manelis 

Academic Editor

PLOS ONE